# Approximate calculations of the net economic impact of global warming mitigation targets under heightened damage estimates

**Patrick T. Brown**[1]*, **Harry Saunders**[2]

**1** Department of Meteorology and Climate Science, San José State University, San José, California, United States of America, **2** Department of Global Ecology, Carnegie Institution for Science, Stanford, California, United States of America

\* patrick.brown@sjsu.edu

## Abstract

Efforts to mitigate global warming are often justified through calculations of the economic damages that may occur absent mitigation. The earliest such damage estimates were speculative mathematical representations, but some more recent studies provide empirical estimates of damages on economic growth that accumulate over time and result in larger damages than those estimated previously. These heightened damage estimates have been used to suggest that limiting global warming this century to 1.5 ˚C avoids tens of trillions of 2010 US$ in damage to gross world product relative to limiting global warming to 2.0 ˚C. However, in order to estimate the *net* effect on gross world product, mitigation costs associated with decarbonizing the world's energy systems must be subtracted from the benefits of avoided damages. Here, we follow previous work to parameterize the aforementioned heightened damage estimates into a schematic global climate-economy model (DICE) so that they can be weighed against mainstream estimates of mitigation costs in a unified framework. We investigate the net effect of mitigation on gross world product through finite time horizons under a spectrum of exogenously defined levels of mitigation stringency. We find that even under heightened damage estimates, the additional mitigation costs of limiting global warming to 1.5 ˚C (relative to 2.0 ˚C) are higher than the additional avoided damages this century under most parameter combinations considered. Specifically, using our central parameter values, limiting global warming to 1.5 ˚C results in a net loss of gross world product of roughly forty trillion US$ relative to 2 ˚C and achieving either 1.5 ˚C or 2.0 ˚C require a net sacrifice of gross world product, relative to a no-mitigation case, though 2100 with a 3%/ year discount rate. However, the benefits of more stringent mitigation accumulate over time and our calculations indicate that stabilizing warming at 1.5 ˚C or 2.0 ˚C by 2100 would eventually confer net benefits of thousands of trillions of US$ in gross world product by 2300. The results emphasize the temporal asymmetry between the costs of mitigation and benefits of avoided damages from climate change and thus the long timeframe for which climate change mitigation investment pays off.

**Data Availability Statement:** The Matlab code used for this analysis is available at https://doi.org/10.5281/zenodo.4002104.

**Funding:** This study was unfunded.

**Competing interests:** The authors have declared that no competing interests exist.

## 1 Introduction

Human economic well-being is affected by the efficiency by which societies convert various inputs (e.g., natural resources, physical capital, human capital, and labor) into goods and services that raise the standard of living of those who consume them. The availability of energy is fundamental to this process, and over the past several centuries humanity has relied heavily on the combustion of fossil fuels to provide this energy. However, the carbon dioxide emitted as a byproduct of fossil fuel combustion alters global biogeochemistry and climate.

The combustion of all available fossil fuels would likely be sufficient to raise global temperatures by more than ~10 ˚C above preindustrial levels [1]. This could trigger a geologically unprecedented climate change that, among a myriad of other consequences, would entail an eventual sea-level rise of over 60 meters [1] and risk a mass extinction [2] that would undoubtedly harm human economic well-being. Reducing all emissions to zero in the very near-term, however, would likely require significant societal disruption which could also have substantial negative consequences for human economic well-being. Given the undesirability of these two extreme cases, it would be rational for humanity to follow an intermediate path to decarbonization [3].

However, the optimal path of decarbonization from a global macroeconomic output perspective (i.e., the emissions reduction rate that maximizes the present discounted value of gross world product), is unlikely to come about under laissez-faire conditions since the economic damages of fossil fuel combustion are external to the private transactions associated with energy acquisition [4–6]. That is, the benefits of fossil fuel combustion are privatized while the costs are socialized both is space and in time. To correct for this market failure, governments have long pursued multilateral agreements to limit global greenhouse gas emissions. In 1992, the United Nations Framework Convention of Climate Change adopted the official objective of stabilizing global temperature at a level that would "avoid dangerous anthropogenic interference with the climate system"[7]. The 2009 Copenhagen Accord defined this global temperature target to be 2 ˚**C** above preindustrial levels [8] and the 2015 Paris Accord enshrined this goal in an internationally legally binding document. The Paris Accord also strengthened the language such that the goal was to remain "well below" 2 ˚**C** and it articulated ambitions for limiting global temperature to 1.5 ˚**C** [9, 10]. However, current national commitments under the Paris Accord are likely to result in global warming closer to 3 ˚**C** above preindustrial levels by 2100 [11].

The most prominent tools used to evaluate the economic implications of various global temperature targets are Integrated Assessment Models (IAMs). There are a wide variety of such IAMs that vary in geographical and sectoral resolution [12], but some of the simplest and most often-used are FUND [13], PAGE [14] and the Dynamic Integrated Climate Economy model (DICE) [15, 16]. These models weigh the benefits of avoided economic damages from climate change against the costs of mitigating greenhouse gas emissions and are often employed to calculate the global greenhouse gas emissions reduction pathway that maximizes the present discounted value of global social welfare. However, in most configurations, these IAMs produce optimal greenhouse gas emissions pathways that result in temperature stabilization levels above the Paris Accord's articulated targets of 1.5 ˚C or 2 ˚C [17]. That is, these IAMs typically calculate that stabilizing temperatures at or below 2 ˚C imposes a global mitigation cost on welfare that is larger than the benefits incurred from avoided damages.

However, the optimal mitigation pathways calculated by IAMs may be reconciled with Paris Accord temperature targets if IAMs either substantially overestimate the cost of mitigation or underestimate the economic damages associated with climate change. Mitigation costs may be overestimated if, for example, induced technological change from climate policy is

underestimated or the rates of cost reductions from learning-by-doing are underestimated [18]. On the economic damages side, a growing body of research has challenged the previous widely-used climate change damage estimates included in these IAMs on the grounds that they neglect important impacts [19, 20], are insufficient in their geographic coverage [20], are insufficient in their extrapolation to high levels of warming [21–23], do not account for synergistic effects [24–26], do not account for environmental tipping points [27–29], do not account for non-substitutability between market and non-market environmental goods [27, 30, 31], and do not account for the impacts on economic growth imposed through influences on the factors of production [21, 32–35].

The representation of economic climate damages in IAMs might be improved by more rigorous grounding in observed relationships between climate conditions and economic output. Indeed, recent research has emphasized the historical/empirical estimation of the economic effects of climate change [36, 37]. In particular, the results of several studies [36–38] suggest substantial effects of temperature change on economic growth. Unlike impacts on the level of economic output in a given year, such impacts on economic growth accumulate over time and can result in substantially higher aggregate estimates of impact than those traditionally calculated in IAMs [35, 39–41]. These computed damages are large enough such that even small differences in global temperature stabilization targets result in large impacts on global gross domestic product (gross world product, GWP).

In particular, Burke at al. [39] found that limiting global warming to 1.5 ˚C relative to 2 ˚C would result in cumulative avoided damages of ~40 trillion 2010 US$ in present discounted value (PDV) of GWP through 2100 at a 3%/year discount rate (we round most GWP values in this paper to the nearest 10 trillion to emphasize the approximate nature of these calculations). As a point of reference, GWP for the single year of 2018 was ~80 trillion US$ [42]. The above ~40 trillion US$ figure is the primary impetus for the analyses conducted in this study. The ~40 trillion US$ number represents the benefit side of the ledger for global warming mitigation of a given level but our goal is to put this number in context by including estimates of the costs associated with remaining below given levels of global warming. Our study follows in the footsteps of several previous studies that incorporated damages to economic growth into DICE [21, 32, 33, 43] but we seek to address the following specific questions. Our primary research question is:

1. If we combine economic damages that emulate Burke at al. [39] (see also [44]) with mainstream mitigation cost estimates included in DICE, what are the *net* effects on GWP associated with achieving the Paris Accord temperature targets?

Auxiliary research questions include:

2. How does the above net effect on GWP change under a spectrum of mitigation stringency levels (and thus a spectrum of levels of global warming by 2100)?

3. How do results compare between the representation of economic damages that emulate Burke at al. [39] and the traditional economic damage representation in DICE?

4. How do results compare as a function of the time horizon considered and the discount rates used?

Throughout this study, we make particular note of results corresponding to the time horizon through 2100 because this has long been used as the standard time horizon considered in the climate change literature. In addition to being used in Burke at al. [39], it is the timeframe used for calculating mitigation costs in Intergovernmental Panel on Climate Change assessment reports [45, 46], and the recent Shared Socioeconomic Pathway studies [47].

In the methods section, we discuss in detail how we modify DICE from its traditional configuration (Section 2.1) and investigate its output under a spectrum of exogenously-defined mitigation stringency levels (Section 2.2 and 2.3). We discuss in detail how we parameterize economic damages that emulate those of Burke at al. [39] into the DICE model (Section 2.4) and calibrate the parameterization (Section 2.4.1). We also discuss the mitigation cost function due to its prominence in our calculations even though it is not modified from its traditional form (Section 2.5). In section 3, we show the results of our analysis from several perspectives and in Section 4 we discuss and conclude.

## 2 Methods

### 2.1 Traditional DICE

The analyses in this study utilize the equations in the Dynamic Integrated Climate-Economy 2016 (DICE2016) model [16]. In its traditional configuration, DICE calculates (using perfect foresight) the time evolution of the emissions control rate (as well as the time-invariant savings rate) that maximizes the present discounted value of total social welfare. Fig 1 provides a schematic of the primary equations in DICE (excluding details on the geophysical equations), as well as their attributes such as whether or not they are predefined outside of the main interactive calculation (exogenous, blue) or calculated at each time-step interactively with the other equations (endogenous, purple).

### 2.2 Exogenous control rate trajectories and the discarding of social welfare

Here, we are interested in using the equations from the traditional DICE model because it ties together mainstream estimates of mitigation costs with economic damages from global warming in a unified framework. However, one of our primary points of focus is on the influence of

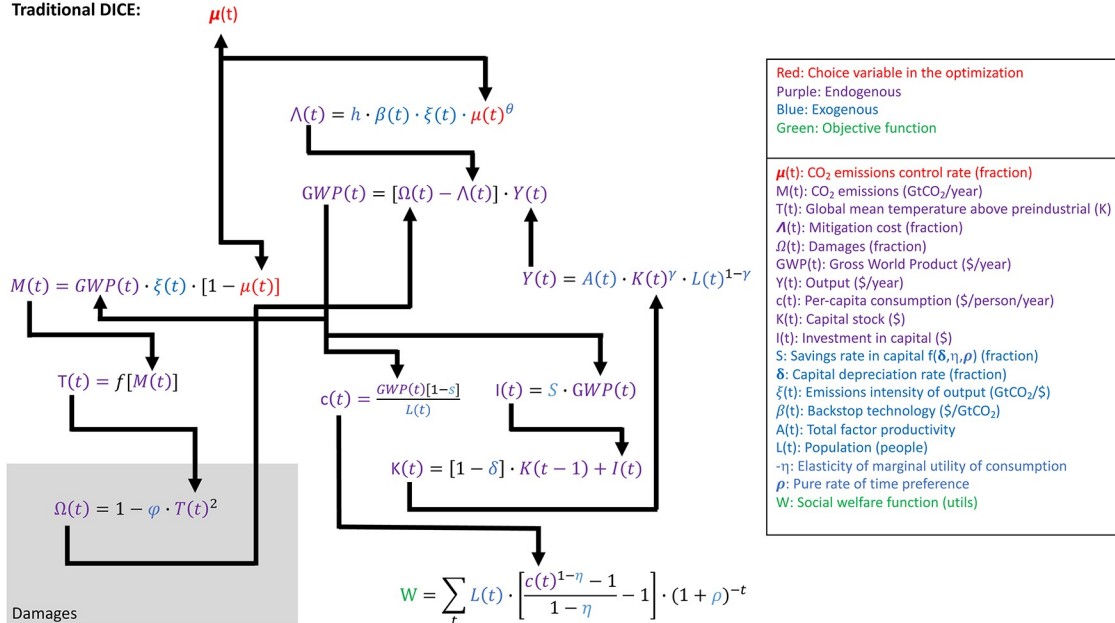

**Fig 1. Primary equations and interactions in the traditional DICE climate-economy model.** Details on the geophysical module of DICE are not shown and are encapsulated in the expression T(t) = f[M(t)]. Variables not defined in the key are constants. See Nordhaus [48] for further details.

various levels of mitigation stringency ($CO_2$ emissions control rates, $\mu(t)$) on GWP as a function of time.

The above goal does not align fully with the purpose of traditional DICE, run in its optimization configuration (Fig 1). In its traditional optimization configuration, normative parameters like the pure rate of time preference and the elasticity of the marginal utility of consumption (which can be thought of as a measure of generational inequality aversion) are used to inform the calculation of the single optimal evolution of $CO_2$ emission control rates, $\mu(t)$ such that total social welfare integrated over an effectively infinite time horizon, is maximized (Fig 1). Even normative choices about whether to optimize for total utility or per-capita utility substantially affect these calculations [49].

We are specifically interested in GWP as a function of time and as a function of the stringency of mitigation; we are not necessarily interested in the single optimal mitigation strategy given a number of normative assumptions. Thus, we eliminate the social welfare function (W in Fig 1) from our framework (Figs 2 and 3) so that the specific values of the normative parameters are not affecting our GWP calculations.

This eliminates the objective function of traditional DICE. We do not simply move the objective function to the discounted present value of GWP because this would incentivize nonsensical behavior in the model like the elimination of consumption of GWP in order to maximize GWP.

Furthermore, we seek a framework that facilitates an investigation of the effects of mitigation as a function of time horizon. If we were to attempt to implement near-term finite time horizons (e.g., through 2050 or 2100) in the traditional DICE framework, the model would

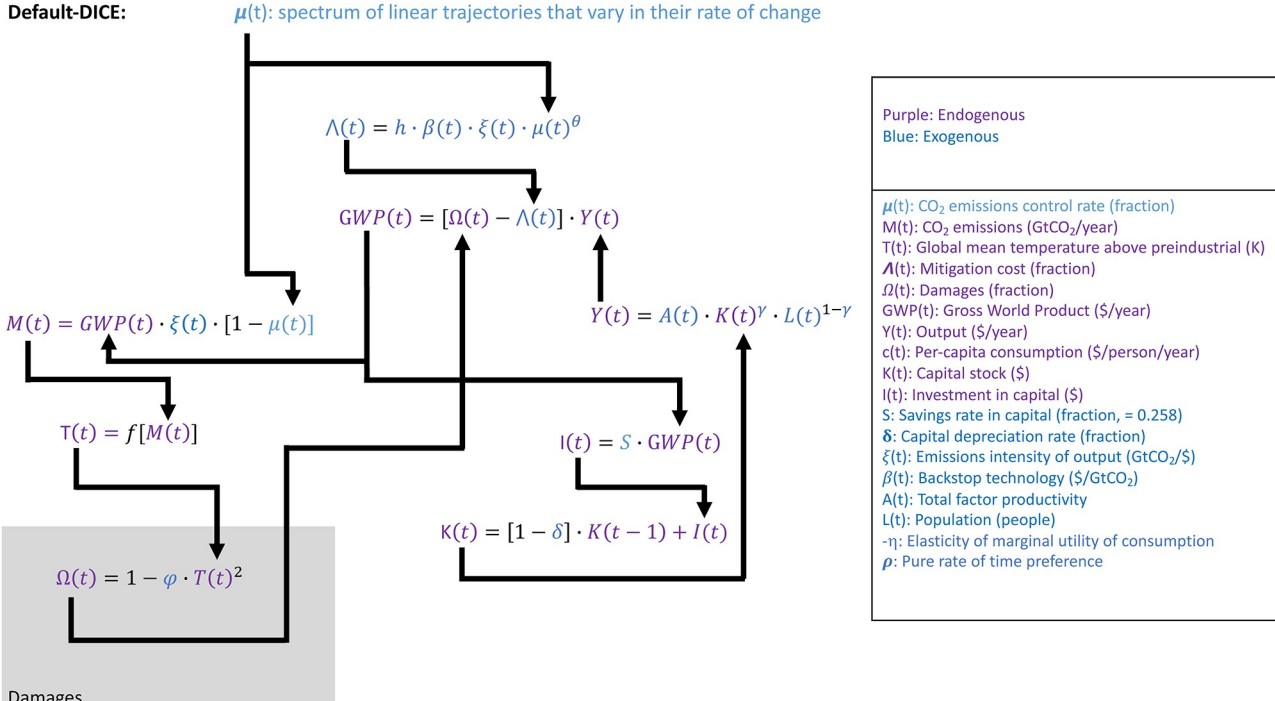

**Fig 2. Primary equations and interactions in the DICE climate-economy model used in this study with the default damage equation.** This version is what is referred to as "Default-DICE" throughout the remainder of the paper. Details on the geophysical module of DICE are not shown and are folded into T(t) = f[M(t)]. Variables not defined in the key are constants. See [48] for further details.

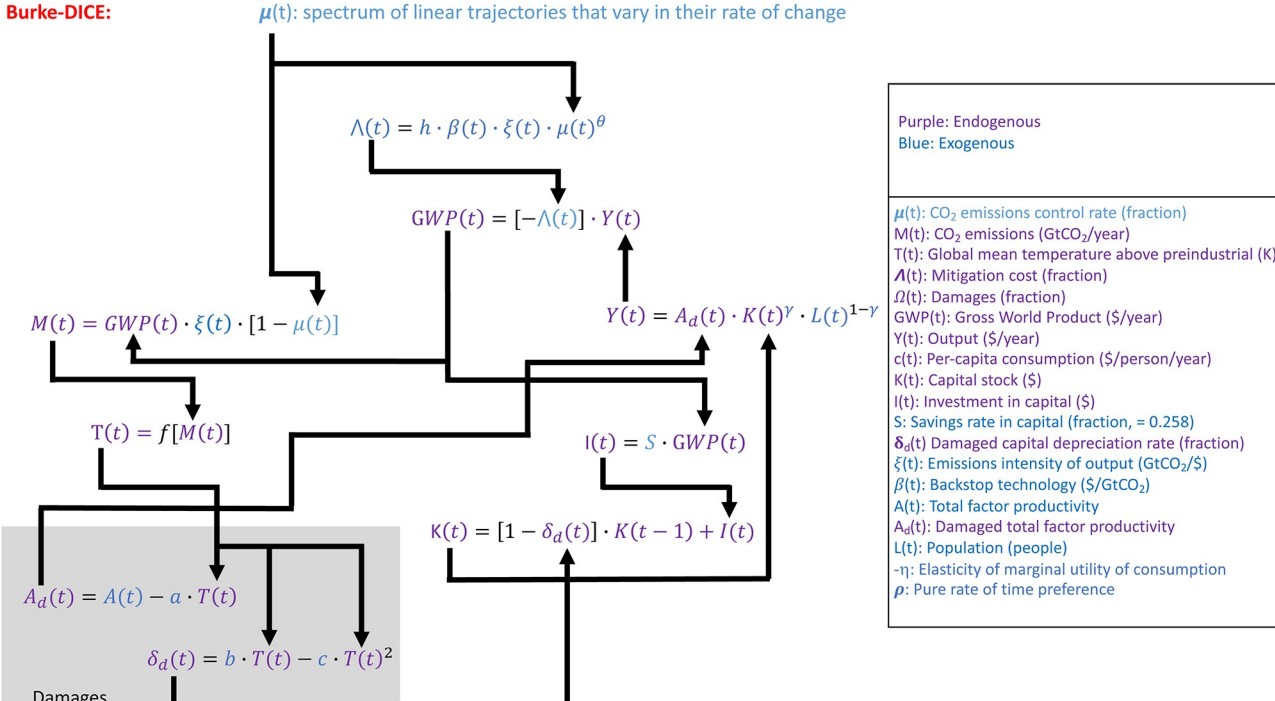

**Fig 3. Primary equations and interactions in the DICE climate-economy model used in this study with the Burke et al. [39] like damages parametrized.** Details on the geophysical module of DICE are not shown and are folded into T(t) = f[M(t)]. Variables not defined in the key are constants. See [48] for further details.

find it optimal to not mitigate climate change at all since the benefits of mitigation would not be fully realizable until after the model's world has ended.

Finally, we are seeking a rough estimate of what might be the practical net effect of various levels of emissions reductions stringencies; we are not necessarily seeking the net effect of the absolute best-case scenario emissions reduction pathway calculated with perfect foresight (what traditional DICE calculates).

Given our goals and the complications mentioned above, we alter DICE such that it is not run in its traditional optimization mode. Instead, we feed DICE a spectrum of greenhouse gas (represented by $CO_2$ alone) emissions reduction control trajectories that vary in their stringency (Figs 2 and 3). This is similar to the methods of previous studies [50]. In this configuration we use a constant savings rate (S) of 25.8%. This savings rate results from the default values of the parameters used in traditional DICE and it is close to historical observations which range from 23.4%-26.8% from 2000–2018 [42].

In our framework, each ensemble of DICE runs is driven by sixty different μ(t) time series that all represent linear increases in $CO_2$ control rates and differ in the timestep (at five year increments) at which 125% control is reached–allowing for net negative emissions (Fig 4). There is also a no-mitigation experiment, where μ(t) = 0 over the entire model run that is used as a baseline for which the other experiments are compared to (Fig 4). Since our exogenously-defined μ(t) timeseries are required to be linear, the μ(t) that we define as "optimal" in this paper is optimal in the sense that it maximized GWP under the constraints of monotonic linear reductions in $CO_2$ emissions. Requiring the μ(t) timeseries to be linear greatly reduces the degrees of freedom in our analysis and thus simplifies our study. Because of these changes, we

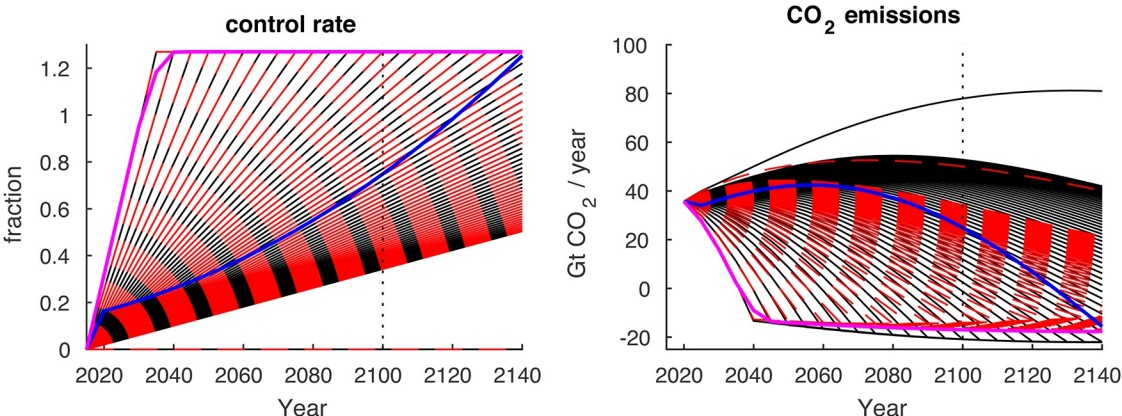

**Fig 4. The spectrum of linearly ramping control rates and their associated $CO_2$ emissions trajectories.** Results from the Default-DICE representation of economic damages are shown with black curves while results from Burke-DICE are shown with red dashed curves. Also shown are two curves overlaying the versions of Burke-DICE (magenta) and Default-DICE (blue) where the control rate is optimized in order to maximize the present discounted value of welfare through 2300 with a 1.5%/year pure rate of time preference. These lines indicate that the control rates calculated under traditional optimization mode are close to linear.

are using equations in DICE to run forward-projections in the same way a typical climate model would be run rather than using the equations to inform the perspective of a benevolent social planner. To summarize, we find that conducting the analysis this way entails three primary advantages:

1. It allows for easy investigation of economic impact as a function of mitigation stringency since it allows us access to DICE economic calculations from emissions reductions pathways that would be considered to be non-optimal under the traditional DICE configuration.

2. It eliminates the influence of normative parameters like the pure rate of time preference, generational inequality aversion and total vs. per-capita utility maximization on our GWP calculations (note that we still time-discount GWP in many calculations of the paper).

3. It removes the inconsistency of discussing results over a finite time horizon (e.g., through 2100) when the emissions reduction pathway was calculated to maximize welfare over a longer time horizon.

Nevertheless, we refer interested readers to Glanemann et al. [43] for an analysis that incorporates the damages of Burke at al. [39] into the traditional optimization DICE framework.

## 2.3 Default-DICE

In this section we discuss the default-DICE representation of climate damages as a preamble to their modification in order to emulate Burke at al. [39] damages.

Climate change is expected to negatively impact global economic output (here measured in 2010 US$ gross world product, GWP) through numerous possible pathways [51, 52] including increased infrastructural damage from more intense cyclones [53] sea level rise [54], decreased crop yields [55, 56], decreased labor productivity [57, 58], increased crime [59, 60], increased energy demand [61, 62], increased human mortality [52, 63] and generally decreased total factor productivity [21, 33].

Default-DICE relates economic impacts of climate change to instantaneous (i.e., in that timestep, t) loss of GWP via a simple quadratic function of global temperature change above preindustrial levels (damages box, Fig 2).

Output [(Y(t) in Fig 2] is calculated with a Cobb-Douglas production function that includes the factors of production of an exogenous total factor productivity A(t), exogenous population L(t) and endogenously calculated physical capital K(t) stock. GWP at each time step is inhibited by the aforementioned damages from climate change [$\Omega(t)$] as well as mitigation costs [$\wedge(t)$].

Thus, under the Default-DICE representation, climate damages are mostly felt instantaneously at each timestep (level effects) and there is little impact of climate change on economic growth (the impact on growth that does occur comes about because lost GWP results in lost investment in K(t)).

## 2.4 Parameterization of Burke et al. [39] damages into DICE

Despite the Default-DICE representation of damages being primarily on levels, several studies [21, 33, 38, 39] suggest that economic damages from climate change are imprinted primarily on the factors of production and thus economic growth. In order to parameterize estimates of the effects of climate change on economic growth in Burke-DICE, we replace the Default-DICE representation of damages expressed in Fig 2 with the procedure of Moore and Diaz [32] and allow global temperature to directly alter the growth rate of total factor productivity [A(t)] and the depreciation rate of physical capital [$\delta(t)$]. We use the same functional forms for these effects as Moore and Diaz [32] (damages box, Figs 3 and 5). We do not allow $\delta(t)_d$ to drop below $\delta(t)$ which is 10%/year. The linear reduction in A(t) with temperature and the nonlinear increase in capital depreciation rate with temperature are not necessarily based on theory but rather a calibration of these variables to the empirically-derived relationship between temperature and growth found in Dell et al. [38] (see the Supplementary Information of More and Diaz [32] for more details).

Conceptually, direct damage on infrastructure (from e.g., more-extreme cyclones or floods) is represented by the enhanced depreciation rate of physical capital with increased global temperature [32, 64] and all other pathways of economic damage (e.g., reduced worker productivity, crop yields, etc.) are represented via a reduction in the background growth rate of total factor productivity with increased temperature [21, 32, 33].

The specific partitioning of impacts between A(t) and $\delta(t)$ is not particularly important for our purposes. Rather, we are primarily concerned with implementing a parameterization that results in damages that are consistent with the associated globally-aggregated damages calculated by Burke at al. [39].

**2.4.1 Validity of the Burke et al. [39] damage estimates.** It should be noted that the severity and validity of the damage estimates from Burke at al. [39] has been disputed [40, 65, 66]. Specifically, Kahn et al. [66] use a different statistical model specification in their historical temperature—gross domestic product regressions and project a much smaller amount of damages per degree of global warming by 2100 than Burke at al. [39] (approximately a 7% loss under a no-mitigation scenarios as opposed to 23% [44]). Letta et al. [65] also use a different statistical model specification than Burke at al. [39] and show a relatively small influence of temperature of economic growth (Total Factor Productivity growth in particular) and only in low-income countries. Finally, Newell et al. [40] show that the best performing statistical models in an out-of-sample test relate temperature to gross domestic product *levels* rather than gross domestic product *growth*. They conclude that the specifications of Burke at al. [39] are

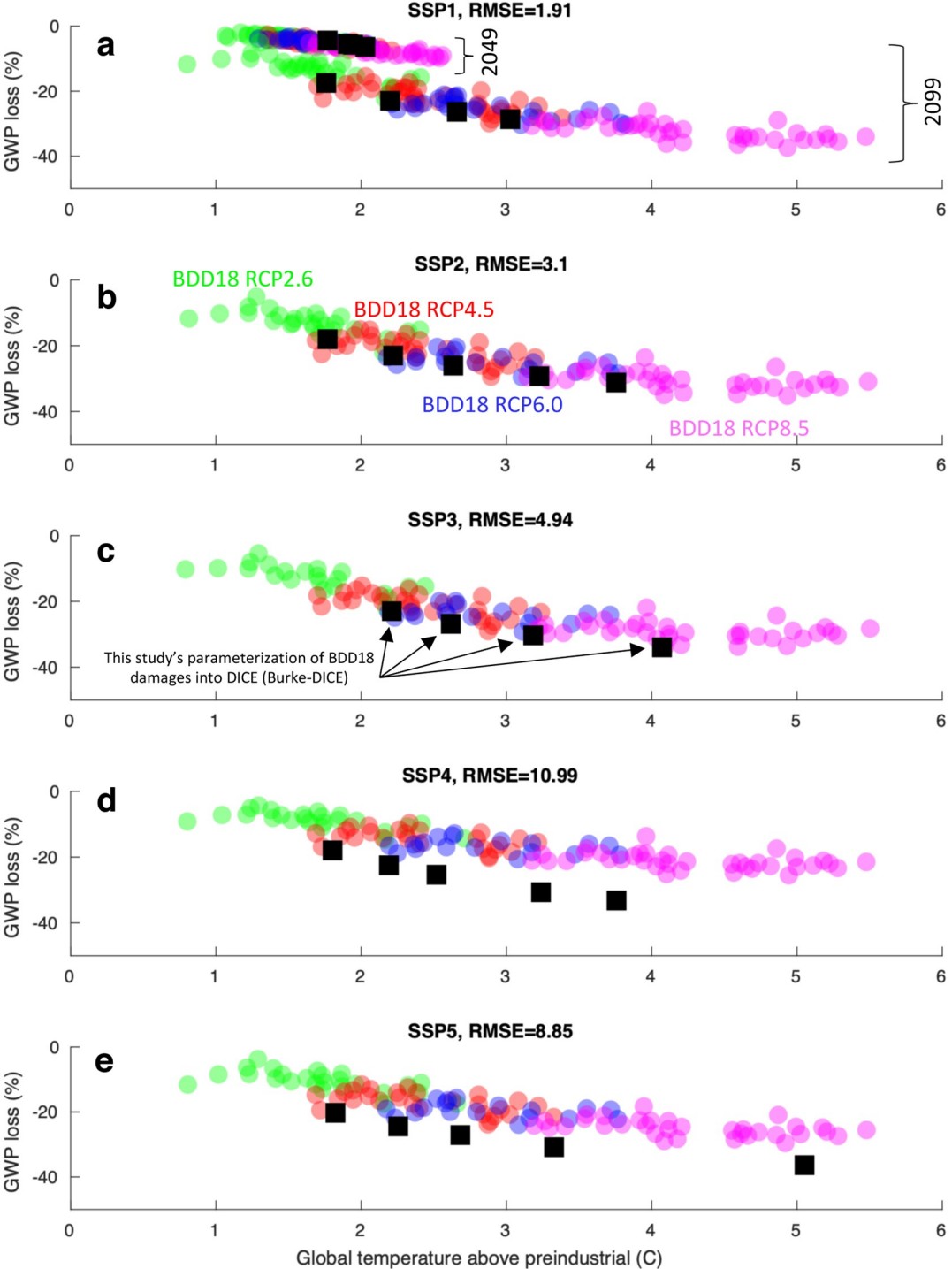

**Fig 5. Economic damage as a function of global temperature above preindustrial levels as calculated in Burke et al. [39] and as calculated by this study's Burke-DICE model.** Colored circles are results directly from Burke et al. [39] (their Fig 4a and Extended Data Fig 6) while black squares are results from this study's Burke-DICE model. The five black squares for each SSP represent five different representative concentration pathways (RCP 2.6, RCP 3.4, RCP 4.5, RCP 6.0, RCP baseline). For Burke-DICE, different Shared Socioeconomic Pathways (SSPs) are represented by substituting the SSPs' population, baseline gross world product (GWP) and global temperature trajectories into the DICE framework. DICE's default configuration is most similar to SSP2 in terms of these parameters and thus this SSP was prioritized in the calibration of Burke-DICE.

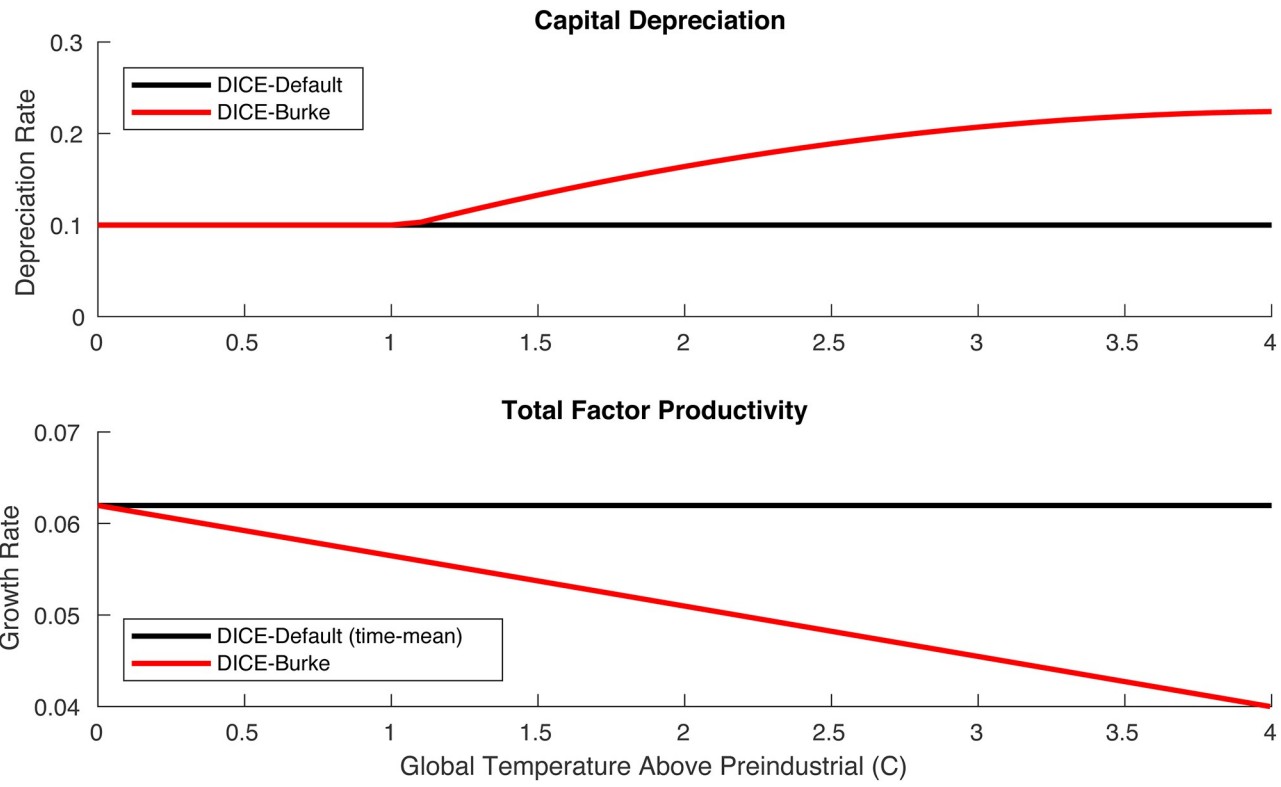

**Fig 6. Capital depreciation rate (per year) and total factor productivity growth rate (per 5 years) in Burke-DICE (red) relative to their default-DICE values (black).** In Burke-DICE, these two parameters are a function of global temperature above preindustrial levels while they are independent of global temperature in Default-DICE.

almost certainly not the optimal specifications and they argue that more accurate specifications amount to economic damages by 2100 of only 1–2%.

Despite these countervailing results, even if Burke at al. [39] overestimate the impact of temperature deviations on gross domestic product historically, it would still be possible for GWP damages to reach the aggregate levels projected by their model by 2100 if the damages come about through pathways other than directly from temperature (e.g., a large sea level rise due to the collapse of the West Antarctic ice sheet).

**2.4.2 Calibration of Burke-DICE parameters.**   Having adopted functional forms by which global temperature can influence GWP growth (damages box in Fig 3), we sought parameter values of 'a', 'b', and 'c' that resulted in GWP damages similar to that reported in Burke at al. [39]. Burke at al. [39] provides estimates of GWP damage under the five Shared Socioeconomic Pathways (SSPs) and five Representative Concentration Pathways (RCPs) (their Fig 4A and Extended Data Fig 6). This provides variation in population, baseline GWP growth, and global temperature (all as a function of time) which should provide sufficient variation to serve as a distributed target for tuning the parameter values of 'a', 'b', and 'c' such that our parameterization roughly replicates the results of Burke at al. [39].

Prior to choosing 'a', 'b', and 'c' parameter values, it was necessary to create analogs to the SSP-RCP combinations within the DICE framework that were consistent with the DICE equations. Towards this end, we combined the global population and baseline GWP trajectories associated with each of the five SSPs with DICE's default elasticity of substitution between labor and capital ($\gamma$) and its default exogenous total factor productivity trajectory A(t). With all

these variables defined, we could solve for the baseline physical capital at each timestep,

$$K(t) = \left( \frac{GWP(t) \cdot L(t)^{\gamma-1}}{A(t)} \right)^{\frac{1}{\gamma}} \qquad (1)$$

and thus solve for the necessary savings rate as a function of time associated with each of the SSPs.

Once the five SSP analogs were created within the DICE framework, their variation in population, baseline GWP, and global temperature (from the RCPs) could be used to tune the 'a', 'b', and 'c' parameter values.

'a', 'b', and 'c' were chosen in a brute-force way (where 'a' values were varied between 0.0001 and 0.01 at 0.0001 increments, 'b' values were varied between 0.05 and 0.50 at 0.001 increments and 'c' values were varied between 0.001 and 0.1 at 0.001 increments. The combination of values that minimized the square error between GWP losses within our framework and those reported in Burke at al. [39] (Fig 5) were used. The resultant values were a = 0.0055, b = 0.105 and c = 0.013. Fig 6 summarizes these results by showing how capital depreciation rate and total factor productivity growth are altered as a function of global temperature in Burke-DICE.

We only use information associated with the SSPs and RCPs for this calibration exercise and we do not use them in our results section which relies instead on exogenous trajectories from DICE2016 [16].

## 2.5 Representation of the costs of mitigating $CO_2$ emissions

We do not modify the default-DICE representation of the costs of mitigation (the $\wedge(t)$ equation is the same in Figs 1, 2 and 3) but we discuss it briefly here due to its prominence as the cost side of the cost-benefit calculation.

DICE models the global aggregate of mitigation costs as an instantaneous (i.e., in that time-step) loss of global output via a simple power function of the fraction of greenhouse gas emissions controlled $\mu(t)$.

$\beta(t)$ represents the larger cost of carbon emission-free energy, like renewable wind and solar energy, relative to the combustion of fossil fuels (or equivalently, the cost of carbon capture and storage and/or atmospheric $CO_2$ removal). $\xi(t)$ accounts for the non-policy induced reduction in the greenhouse gas emissions intensity of the economy through natural increases in energy efficiency (e.g., via improved technology or a transition to a more service-oriented economy) and increases in the fraction of primary energy produced from non-carbon emitting sources. The fraction of greenhouse gas emissions controlled [$\mu(t)$] is the exogenous driver of mitigation effort in our configuration (Fig 4). The convexity parameter $\theta > 1$ represents the notion that the expense of marginal emissions reductions increases with the fraction of emissions abated [67].

Although this representation of mitigation cost is highly idealized, it produces results similar to that of disaggregated process-based IAMs (Fig 7) that simulate the situation in a more sophisticated manner by explicitly representing e.g., a full energy technology portfolio, cost reduction through learning, technology diffusion rates, regional disaggregation, capital costs, etc. [68–71]. This consistency results because the mitigation cost function in DICE was calibrated against these more sophisticated models [72].

Despite the consistency between DICE's mitigation cost representation and those calculated from more sophisticated models, there remains substantial uncertainty associated with all of the terms in the mitigation cost equation as well as in other terms in the DICE model. Thus, in Section 3.3 we discuss the results produced from a set of 2,000 Monte-Carlo trials where the

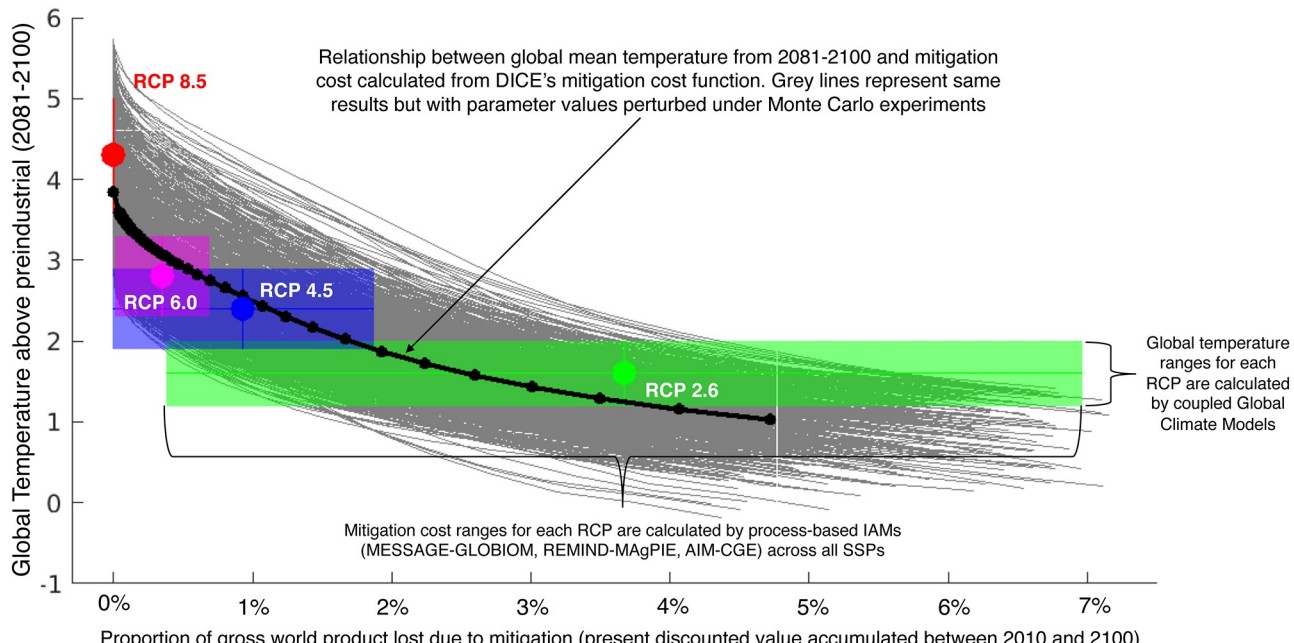

**Fig 7. Comparison of mitigation costs (as a function of level of mitigation effort, represented by global temperature above preindustrial values realized at the end of the 21$^{st}$ century) between DICE's calculations (grey and black lines) and calculations from disaggregated, process-based IAMs [47].** This Figure indicates that DICE calculates similar levels of mitigation costs as more-sophisticated process-based IAMs which is consistent with DICE being calibrated to results from these more-sophisticated IAMs [72]. The data ranges were obtained from Bindoff et al. [73] for global temperature and from S2 Fig in Riahi et al. [47] for present discounted value of gross world product. The discount rate used in these cases is 5%/year.

values of eight parameters are perturbed between two-thirds and three-halves of their default values. Included among these varied parameters are $\beta$, $\xi$, and $\theta$ from the mitigation cost function (grey lines in Fig 7). Overall, we find that mitigation costs in DICE are consistent with more sophisticated calculations and represent a rough but mainstream estimation.

## 3 Results

Our focus here is on how GWP is influenced by the level of mitigation stringency and thus the level of global warming in 2100. The spectrum of various levels of mitigation effort and their associated outcomes are represented by the fanning curves in Fig 8, where Default-DICE results are shown with black curves and Burke-DICE results with red dashed curves.

Our no-mitigation baseline scenarios result in peak emissions between about 50 and 80 GtCO$_2$/year (Fig 8a) and global warming of between 3.5 ˚C and 4 ˚C by 2100 (Fig 8b), roughly consistent with the no-mitigation baseline scenarios of Shared Socioeconomic Pathways (SSPs) 2, 3 and 4 that are calculated by process-based IAMs [47]. Note that the difference in no-mitigation peak emissions between Default-DICE and Burke-DICE results from Burke-DICE's larger and compounding damages feeding back on economic production and thereby reducing CO$_2$ emissions [74]. In our most stringent mitigation case, we allow CO$_2$ emissions to cross zero and become net negative by 2040 –similar to the most ambitions decarbonization pathways envisioned in the SSPs [75, 76] (Fig 8a, green shading).

Below, we adopt the convention of identifying (i.e., labeling) the stringency of mitigation with the level of global warming realized in 2100 (i.e., where Default-DICE and Burke-DICE trajectories cross the vertical dotted line in Fig 8b). Thus, the more stringent the mitigation, the lower the temperature in 2100.

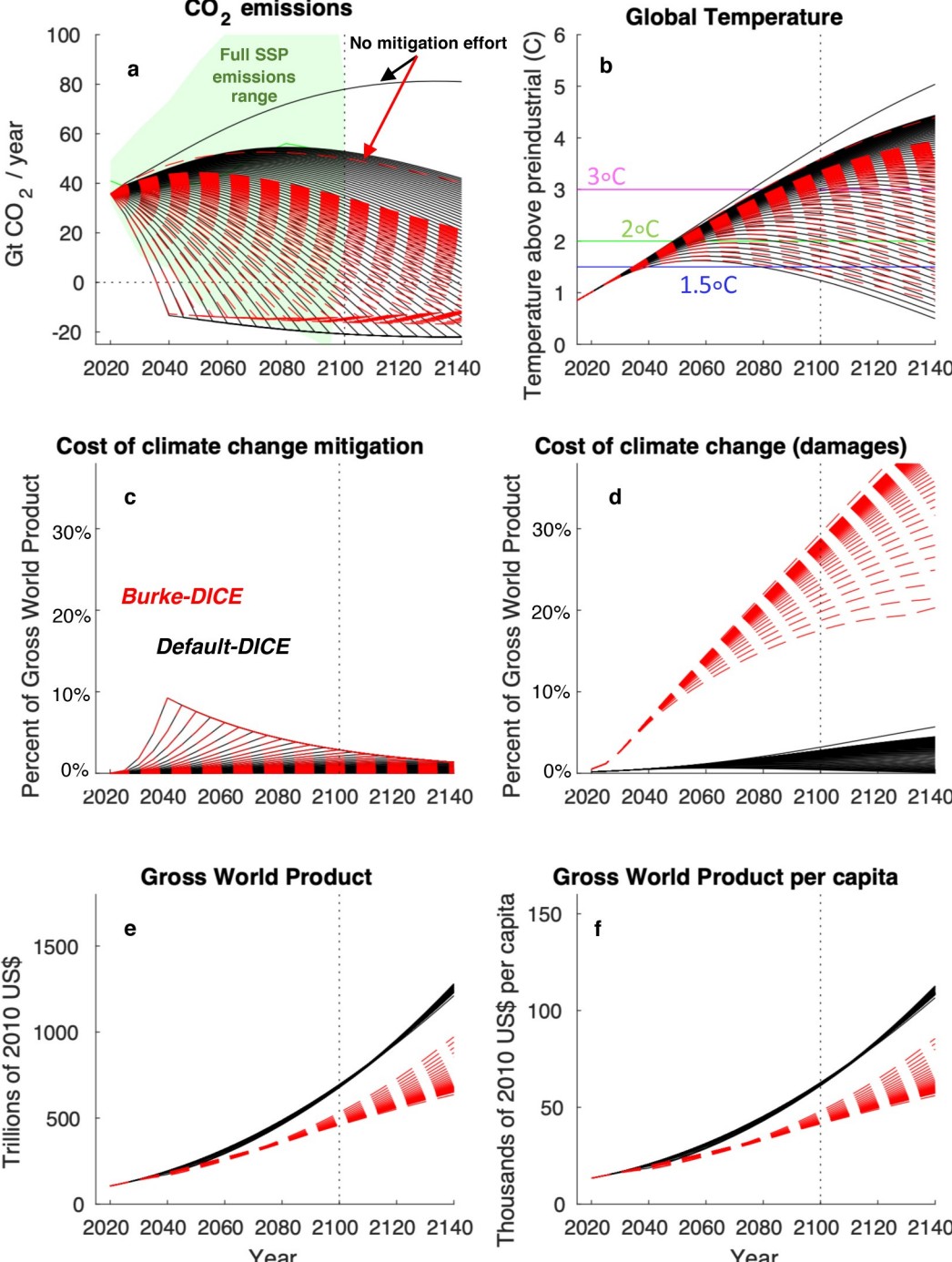

**Fig 8. All model runs conducted, showing the full spectrum of various levels of mitigation effort as well as the difference between default and heightened economic damages from climate change.** Results from the Default-DICE representation of economic damages are shown with black curves while results from Burke-DICE are shown with red dashed curves. Each line represents a different level of mitigation stringency. The green shading in (a) displays the full range of possible future $CO_2$ emissions calculated by processed-based Integrated Assessment Models for the Shared Socioeconomic Pathways (SSPs, including the RCP1.9 scenario [47]), showing that our range samples the most stringent mitigation pathways envisioned in these scenarios.

### 3.1 Costs and benefits of mitigation through time

In addition to $CO_2$ emissions and global temperature, Fig 8 also shows economic outcomes under each level of mitigation stringency. As might be expected, the largest difference between Burke-DICE and Default-DICE is seen in the calculation of the economic cost of climate change in terms of the fraction of global GWP lost (Fig 8d), with Burke-DICE damages resulting in much greater losses of GWP than Default-DICE damages. Under Burke-DICE damages, ~15% to ~30% of GWP is lost in 2100 (relative to a counterfactual of no climate change). By design, this is consistent with the range calculated in Burke at al. [44]. This is in stark contrast to the Default-DICE calculation which shows about one-half to three percent of GWP is lost in 2100 depending on the global temperature. Also, since the Burke-DICE damages are levied on factors of production (as opposed to GWP "levels" as is the case for Default-DICE), economic damages under Burke-DICE continue to accumulate even after global temperature crests and begins to decrease (note the monotonic increase in the fraction of GWP lost even under the most stringent mitigation efforts for Burke-DICE, Fig 8d).

Mitigation costs as a fraction of GWP, which are the same for Default-DICE and Burke-DICE, are shown in Fig 8c. DICE's representation of mitigation cost is highly idealized but produces global results consistent with those calculated from disaggregated, process-based IAMs [72] (Fig 7). Achieving the most stringent temperature stabilization targets entails the highest cost primarily because carbon-free energy sources are expected to be more expensive in the near term. In particular, DICE calculates that the mitigation costs associated with limiting global temperatures to below 1.5 ˚C are associated with a reduction of nearly 10% of GWP per year in the 2040s for both Default-DICE and Burke-DICE (relative to the no-mitigation case; Fig 8C).

In the DICE framework, economic growth is projected to be strong enough to outstrip combined climate change mitigation costs and damages in all our model runs: all trajectories have substantial increases in GWP through time (Fig 8e and 8f). Thus, in order to highlight the effect of various levels of mitigation effort, the shading in Fig 9 shows differences in per-capita GWP relative to the no-mitigation case. Negative values (red shading) indicate that the given level of mitigation effort (labeled by the level of global warming in 2100 displayed on the y-axis) results in economic losses relative to the no-mitigation case and positive values (green shading) indicate that the given level of mitigation effort results in economic gains relative to the no-mitigation case (Fig 9). The solid black curves separate positive from negative values and thus delineate the year at which the given level of mitigation effort results in higher per-capita gross world product than the no-mitigation case (which we refer to as the break-even year).

Under both representations of damages, mitigation reduces per-capita GWP in the near-term but increases per-capita GWP in the long-term. The more stringent the level of mitigation (moving towards the bottom of the Figures), the more economic loss in the near term and the larger the economic gain in the long term (Fig 9). Thus, all levels of mitigation considered in this framework eventually confer a net economic benefit with the magnitude of the long-term benefit increasing with the level of mitigation effort.

Since Burke-DICE damages (Fig 9b) are substantially larger than Default-DICE damages (Fig 9a), they produce a shorter payback period (the break-even year is sooner in Burke-DICE than in Default-DICE). This effect is particularly strong at the higher levels of mitigation effort. Under the Default-DICE damage representation, the mitigation effort necessary to limit global warming to below approximately 3 ˚C is associated with a payback period that extends into the 22nd century (Fig 9a). Under Burke-DICE damages, on the other hand, net economic gains from any mitigation begin to be realized within the 21st century (Fig 9b). For more ambitious

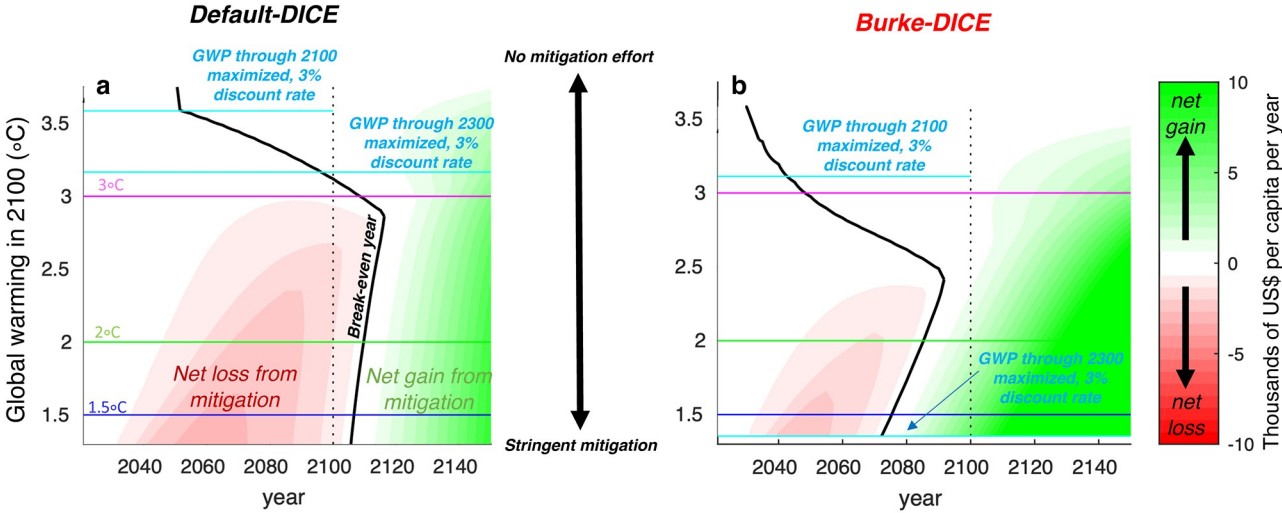

**Fig 9. Effect of the level of mitigation effort on per-capita gross world product through time for both default-DICE and Burke-DICE representations of damages from climate change.** Plots contour the difference in per-capita gross world product between the no-mitigation scenario and the mitigation scenario which results in the global warming above preindustrial levels labeled on the y-axes (in 2100). Red shading indicates that the net effect of mitigation is to cause a loss in that year and green shading indicates that the net effect of mitigation is to cause a gain in that year. As avoided damages increase over time, mitigation effort eventually "breaks-even" (black curves). Light blue horizontal lines identify the level of mitigation effort that maximizes the time-discounted (3%/year) per-capita gross world product through 2100 and through 2300. The effect of heightened damages (Burke-DICE, panel b) is to decrease the net loss from mitigation in the near term, increase the net gain in the long term and to move the break-even year backward in time.

1.5 ˚C and 2 ˚C temperature targets, Default-DICE damages imply a 21[st] century entirely characterized, by GWP sacrifice in favor of future generations while Burke-DICE damages imply that net global mean benefits will begin to be realized by the 2070s to 2080s, within the lifetimes of many people alive today.

Although mitigation effort eventually results in net benefit in all cases, the total economic impact of a given level of mitigation is typically quantified with a present discounted value (PDV) metric under which future economic outcomes are time-discounted and accumulated. This practice is typically implemented under a framework that considers climate change mitigation to be an investment analogous to any generic financial investment. Under such a framework, it is not sufficient to justify a given mitigation effort on the grounds that it will eventually confer a net benefit. In addition, it should be demonstrated that the mitigation effort out-performs reasonable alternative investments (in e.g., education, healthcare, or the direct alleviation of poverty) and thus it is worth not only the absolute cost but also the opportunity-cost of foregone investment elsewhere [77]. This notion is expressed mathematically by discounting the net economic effect of the mitigation effort, typically at an annually compounding rate of several percent per year (the discount rate, r), emulating the prevailing market interest rate in alternative investments [78]:

$$Influence\ of\ mitigation\ on\ the\ PDV\ of\ GWP\ =\ \sum\nolimits_{t\,=\,0}^{time\ horizon} \frac{GWP(t)_{mitigation} - GWP(t)_{no\ mitigation}}{(1+r)^t} \quad (2)$$

Fig 9 also shows the level of mitigation effort (out of our spectrum) that maximizes the per-capita GWP, under 3%/year discount rates (r = 0.03), for time horizons through 2100 and 2300 (horizontal light blue lines). Here we apply a 3%/year discount rate to be comparable with the central calculations of Burke at al. [39], but we test the sensitivity to different discount rates in the following sections. We make note of results corresponding to the time horizon

through 2100 because this has long been used as the standard time horizon considered in the climate change literature. It is the timeframe used for calculating mitigation costs in Intergovernmental Panel on Climate Change assessment reports [45, 46], recent Shared Socioeconomic Pathway studies [47], and in Burke at al. [39].

Under a discount rate of 3%/year and a time horizon through 2100 (the central values used in Burke at al. [39]), the level of mitigation effort the maximizes per-capita GWP shifts from ~3.6 ˚C under Default-DICE to ~3.2 ˚C under Burke-DICE (Fig 9). Remember here that we are not considering benefits beyond 2100 so these values are not comparable to previously-published optimal temperature trajectories using traditional DICE. Thus, even under heightened damages emulating from Burke at al. [39], the 2 ˚C and 1.5 ˚C targets outlined by the Paris Accord imply larger mitigation costs than benefits from avoided damages in terms of the present discounted value of per-capita GWP through 2100. Specifically, we calculate that achieving the 1.5 ˚C and 2.0 ˚C targets result in GWP losses of ~100 trillion US$ and ~60 trillion US$ respectively relative to the no-mitigation case in Burke-DICE (Fig 10a). This implies a ~40 trillion US$ *loss* from limiting global warming to 1.5 ˚C relative to 2.0 ˚C under Burke-DICE (Fig 10a) which is in contrast to the corresponding calculation in Burke at al. [39] which found a central estimate of a ~40 trillion US$ benefit from limiting global warming to 1.5 ˚C relative to 2.0 ˚C but did not consider mitigation costs.

Additionally, as discussed in Section 2.4.1, the magnitude of the Burke at al. [39] damages to economic growth have been challenged, with other groups obtaining results that do imply damages to growth but damages that are less severe and result in less aggregate impact by 2100. One example is Kahn et al. [66] who find damages to growth that would result in a 7% loss of per-capita GWP by 2100 in a no-mitigation case and a 1% loss of per-capita GWP by 2100 under mitigation that results in limiting global warming to 2 ˚C.

In order to investigate the net costs associated with damages such as these, we re-tuned the DICE-Burke model (Fig 3) such that the accumulated damages on growth matched those found in Kahn et al. [66]. Specifically, we re-tuned the coefficients of the model such that in the no-mitigation case, damages on per-capita GWP in 2100 matched the 7% projected by Kahn et al. [66] and in the 2 ˚C case, damages on per-capita GWP in 2100 matched the 1% projected by Kahn et al. [66]. ('a' was reduced to 1% of its Burke-DICE value, 'b' was reduced to 50% of its Burke-DICE, 'c' was reduced to 10% of its Burke-DICE and an exponent of 2 was added to the temperature term in the damages to total factor productivity).

Using the above damage estimates, we calculate that achieving the 1.5 ˚C and 2.0 ˚C targets result in net GWP losses of ~160 trillion US$ and ~60 trillion US$ respectively, relative to the no-mitigation case through 2100 (which is in between those of Default-DICE and Burke-DICE, Fig 10). This implies a ~100 trillion US$ loss from limiting global warming to 1.5 ˚C relative to 2.0 ˚C which is also in between the ~40 trillion US$ loss calculated under Burke-DICE and the ~120 trillion US$ loss under Default-DICE (Fig 10).

To more directly compare our Buke-DICE mitigation cost and damage calculations to existing mitigation cost estimates and to the damage estimates from Burke at al. [39], we impose our fractional mitigation cost and damage trajectories for the 1.5 ˚C and 2.0 ˚C targets (Fig 8c and 8d) on the baseline SSP2 GWP trajectory [47].

Under Burke-DICE, we find that limiting global warming to 1.5 ˚C costs ~220 trillion US$ in GWP and that limiting global warming to 2.0 ˚C costs ~120 trillion US$ in GWP indicating that it costs an additional ~100 trillion US$ to move from 2.0 ˚C to 1.5 ˚C (3%/year discount rate). This is comparable to the representative marker scenario for SSP2 calculated by the MESSAGE-GLOBIOM IAM [79] which shows that limiting global warming to 1.5 ˚C costs ~190 trillion US$ and that limiting global warming to 2.0 ˚C costs ~70 trillion US$, suggesting that it costs an additional ~120 trillion US$ to move from 2.0 ˚C to 1.5 ˚C (3%/year discount

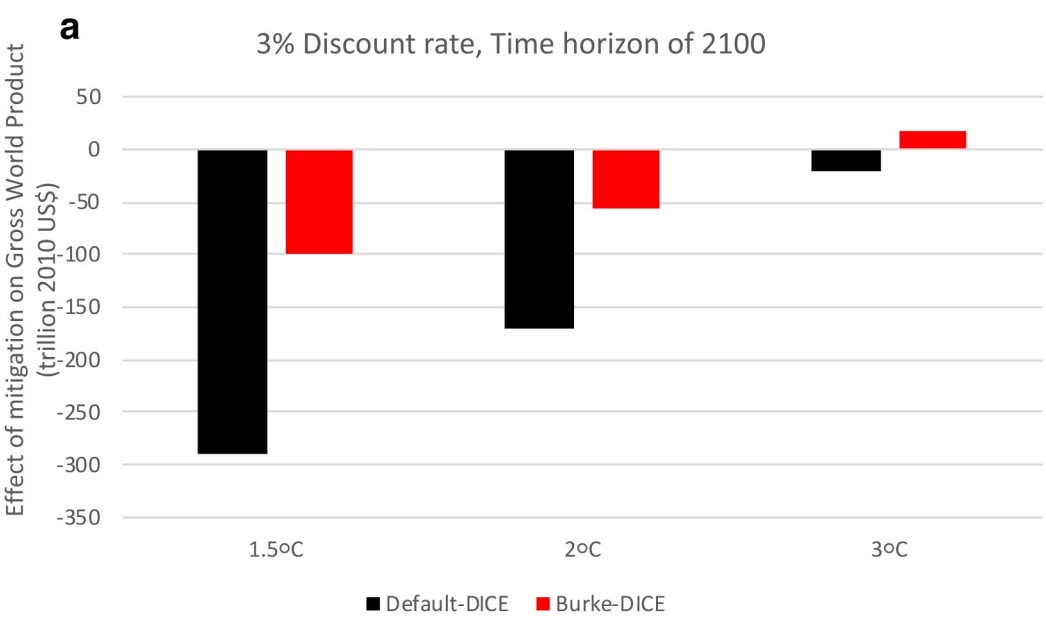

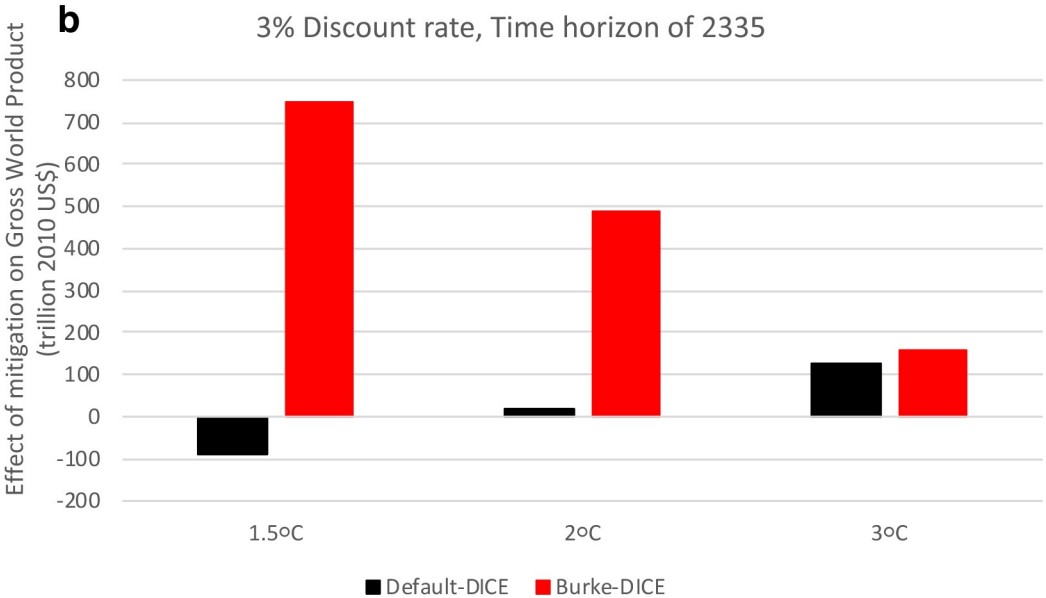

**Fig 10. Net economic benefit (or loss) of three temperature targets (1.5 ˚C, 2 ˚C and 3 ˚C in 2100) for both default and heightened economic damages from climate change.** Values are the difference in gross world product (in terms of present discounted value, time-discounted at 3%/year) between the given level of mitigation effort and the no-mitigation scenario. Under Burke-DICE damages, limiting global warming to 1.5 ˚C results in a net loss of gross world product relative to the no-mitigation case and relative to the 2 ˚C level of mitigation, for a time horizon of 2100 (red bars in a). However, for a time horizon of 2300 under Burke-DICE damages (red bars in b) limiting global warming to 1.5 ˚C results in a large net gain in gross world product relative to the no-mitigation case and relative to the 2 ˚C level of mitigation.

rate). All of the above numbers fall within the range of a recent meta-model study of mitigation costs [80] (when the discount rate is adjusted to 5%/year to match the meta-model). In particular, at a 5%/year discount rate, we calculate that limiting global warming to 1.5 ˚C costs ~90 trillion US$ and that limiting global warming to 2.0 ˚C costs ~40 trillion US$ which are within

the 2σ confidence intervals of 10 to 104 trillion US$ for 1.5 ˚C and 4 to 63 trillion US$ for 2.0 ˚C [80].

Under Burke-DICE, we find that limiting global warming to 1.5 ˚C amounts to damages of ~520 trillion US$ and that limiting global warming to 2.0 ˚C amounts to damages of ~600 trillion US$, indicating that there is an ~80 trillion US$ benefit of avoided damages from moving from 2.0 ˚C to 1.5 ˚C (3%/year discount rate). This is within the 2σ confidence interval of Burke at al. [39] which was -54 to 101 trillion US$ (their Extended Data Table 1) but not precisely on the median because our model was calibrated to match Burke at al. [39] over an extensive parameter-space (Fig 5) and did not target one single damage estimate.

Since mitigation costs dominate in the near term and damages dominate in the long term, shorter time horizons disproportionally weigh and sample the period associated with economic losses and thus they push up the per-capita GWP maximizing temperature target. If the time horizon considered is extended to 2300, the optimal level of mitigation effort shifts from ~3.6 ˚C to ~3.2 ˚C under Default-DICE but it shifts from ~3.2 ˚C to below 1.5 ˚C under Burke-DICE, in this case justifying the most stringent Paris Accord target from a GWP perspective (Fig 9). Under a time horizon of 2300 (and a 3%/year discount rate), Burke-DICE damages suggest that achieving the 1.5 ˚C and 2.0 ˚C targets would result in GWP gains of ~750 trillion US$ and ~500 trillion US$ respectively, relative to the no-mitigation case (Fig 10b). This implies a gain of ~250 trillion US$ from limiting global warming to 1.5 ˚C relative to 2.0 ˚C under Burke-DICE (Fig 10b).

## 3.2 Conditions where Paris Accord temperature targets maximize per-capita gross world product

In addition to the time horizon, the influence of the discount rate on the level of mitigation effort that maximizes per-capita GWP is particularly relevant since these two parameters are largely subjective and yet strongly influence the calculations. Fig 11 shows the combined effect of discount rate and time horizon on the temperature value that maximizes GWP under the two different representations of damages. Here the discount rate is applied directly to the GWP(t) time series (Eq 2) and thus it is not the same thing as the pure rate of time preference in traditional DICE (ρ in Fig 1). Under Default-DICE damages, the 2 ˚C target does not maximize GWP unless a time horizon of greater than 2160 at a 0% discount rate is considered

**Table 1.  DICE variables whose values were randomly perturbed in the Monte Carlo trials described in the text and the range over which they were perturbed.**

| Variable | Range over which values are perturbed in Monte Carlo experiments (all are between 2/3 and 3/2 of their default value) |
|---|---|
| mitigation cost curve exponent, $\theta$ | 1.73–3.90 (unitless) |
| $2 \times CO_2$ climate sensitivity, informs T(t) = f [M(t)] | 2.07–4.65 (∘C) |
| Initial growth rate for total factor productivity per 5 years, informs A(t) | 5.07–11.4 (% per 5 year) |
| $CO_2$ intensity of economy growth rate, informs ξ(t) | -1.01--2.28 (% per year) |
| Decline rate of total factor productivity per 5 years, informs A(t) | 0.33–0.75 (% per 5 years) |
| asymptotic population level, informs L(t) | 7.667–17.250 (billion people) |
| backstop cost decline rate, β(t) | 1.67–3.75 (% per 5 years) |
| $CO_2$ intensity of economy, change in growth rate, informs ξ(t) | -0.07--0.15 (% per 5 years) |

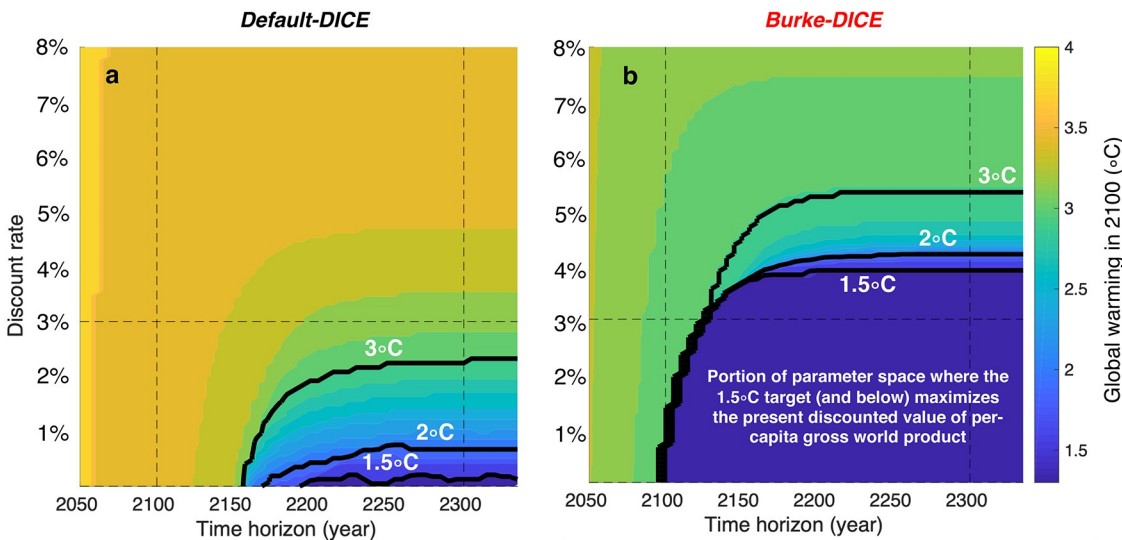

**Fig 11. Sensitivity of the optimal level of mitigation effort to the discount rate and time horizon for both default-DICE and Burke-DICE economic damages from climate change.** The level of mitigation effort that maximizes the present discounted value of time-discounted per-capita gross world product is labeled by the associated level of global warming in 2100. The time horizon is the end year of each calculation where the beginning year is 2020 in all cases. Considering heightened damages (b) causes the 2 ˚C and 1.5 ˚C targets to maximize gross world product under a much more expansive combination of discount rates and time-horizons.

(Fig 11a) or a time horizon of ~2250 at a ~0.6% discount rate is considered. The 1.5 ˚C target does not maximize GWP under Default-DICE unless time horizons of beyond 2200 are considered in conjunction with discount rates below ~0.25%. In contrast, under Burke-DICE damages, much more of the time horizon/discount rate parameter space justifies the Paris Accord temperature targets from a GWP perspective. Under these heightened damages, time horizons of greater than ~2150 and discount rates lower than ~4% result in GWP-maximizing levels of mitigation at or below 2 ˚C; and time horizons of greater than ~2175 and discount rates of less than ~3.8% result in GWP-maximizing temperature stabilization levels below 1.5 ˚C.

Under higher discount rates that roughly match the recent historical real return on U.S. capital (e.g., ~7%/year [77]), even Burke-DICE damages do not justify much climate change mitigation from a GWP perspective (Fig 11b). This highlights the power of supposing indefinite exponential economic growth. Specifically, this result suggests that if we suppose that alternative investments in capital, education and technology would yield returns of >7%/year indefinitely, without any hindrance due to climate change, then these alternative investments would be preferable to climate mitigation, even under Burke-DICE damages. However, on a finite planet that utilizes labor, energy and natural capital for production, we cannot expect exponential economic growth indefinitely, particularly in the face of unmitigated climate change. In particular, it may be more appropriate to think of GWP as being on a logistic trajectory than an exponential one. This uncertainty in future return on investment justifies the use of discount rates that declines over time [81], though we do not investigate such a discounting framework here.

## 3.3 Sensitivity of net cost calculations

In addition to investigating the sensitivity of the optimal level of mitigation effort to the time horizon and discount rate (Fig 11), we also investigate the influence of other DICE parameter

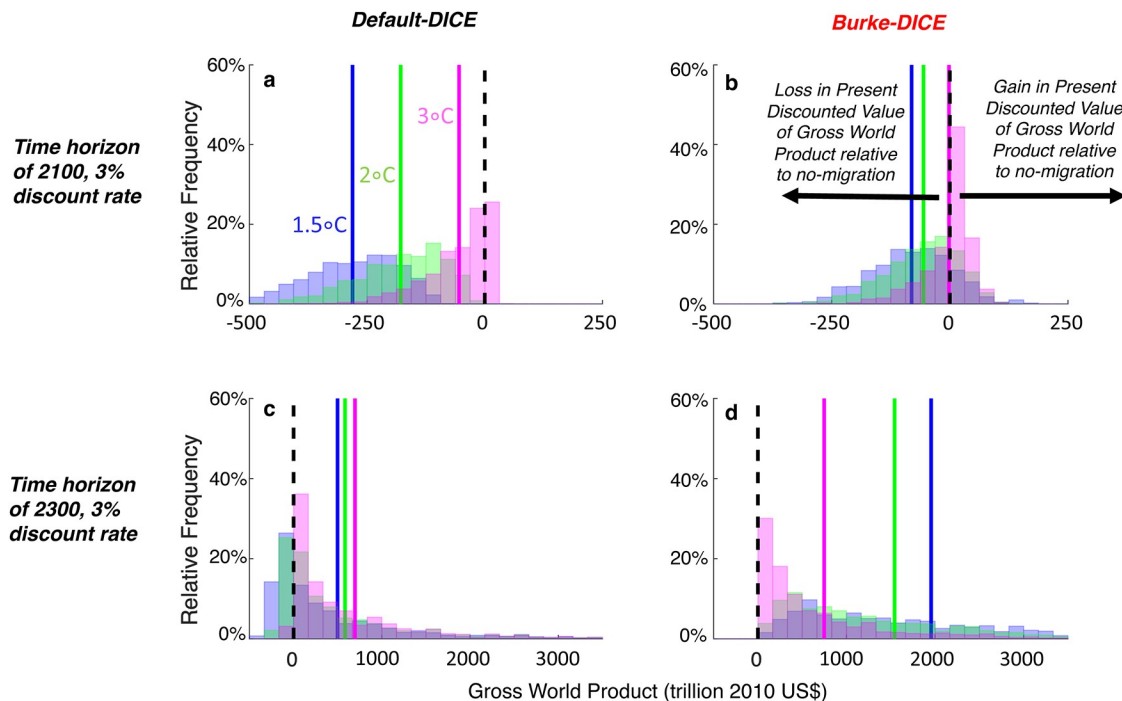

**Fig 12. Net economic benefit (or loss) of three temperature targets (1.5 ˚C, 2 ˚C and 3 ˚C in 2100) for both default and heightened economic damages from climate change.** Histograms plot the distribution, across Monte Carlo trials, of the difference in gross world product (in terms of present discounted value, time-discounted at 3%/year) between the scenario closest to achieving the given temperature target and the no-mitigation scenario. Two thousand Monte Carlo trials were performed in which eight geophysical and socioeconomic DICE parameter values were perturbed between two-thirds and three-halves of their default values (Table 1). In the four scenarios considered here, the 1.5 ˚C level of mitigation only tends to be economically superior to the 2 ˚C level of mitigation under Burke-DICE damages and a time horizon of 2300 (panel d, where the blue line is to the right of the green line).

values that represent socioeconomic and geophysical phenomena with substantial uncertainty [72]. Specifically, we conduct 2,000 Monte Carlo trials that perturb eight parameters of interest within a range of two thirds to three halves of each parameter's default value (Table 1). In each of the 2,000 trials, all eight variables had their default value multiplied by one of the following coefficients where all values were equally likely: 0.6667, 0.7143, 0.7692, 0.8333, 0.9091, 1.0000, 1.1000, 1.2000, 1.3000, 1.4000, 1.5000. The goal of this exercise is to obtain a first-order estimate of the sensitivity of our calculations to the values of these parameters.

Fig 12 shows the distributions across Monte Carlo trials of the net economic impact of the three levels of mitigation (1.5 ˚C, 2 ˚C and 3 ˚C of global warming in 2100) relative to the no-mitigation case. Under a time horizon of 2100, the net economic effect of the 3.0 ˚C mitigation level is more positive than the 2.0 ˚C level, which in turn is more-positive than the 1.5 ˚C level under both Default-DICE and Burke-DICE damages though there is substantial overlap in the distributions (Fig 12a and 12b).

For Default-DICE, the 1.5 ˚C and 2.0 ˚C temperature targets result in economic losses through 2100 in every Monte Carlo trial (Fig 12a). For Burke-DICE, the 1.5 ˚C and 2.0 ˚C levels of mitigation have the majority of their distributions on the negative side of the ledger (~80% and ~70%, respectively; Fig 12b). Through 2100, GWP loss tends to be larger under 1.5 ˚C than it is under 2.0 ˚C under both damage representations (Fig 12a and 12b). This is the case in every Monte Carlo trial for Default-DICE (Fig 13a, black) and ~85% of the trials under Burke-DICE (Fig 13a, red). These results indicates that even under the heightened damages of Burke-DICE, limiting global warming to 1.5 ˚C results in a net loss of GWP relative to 2 ˚C

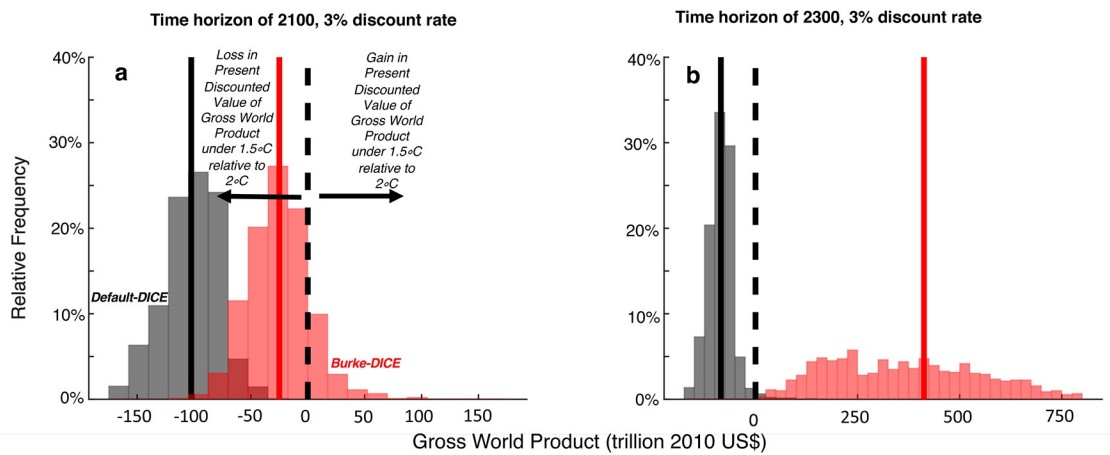

**Fig 13. Difference in gross world product between the 1.5 ˚C and 2.0 ˚C levels of mitigation effort.** Histograms plot the distribution, across Monte Carlo trials, of the difference in gross world product (in terms of present discounted value, time-discounted at 3%/year) for the time horizon of 2100 (a) and 2300 (b) and for both Default-DICE (black) and Burke-DICE (red). Negative values indicate a loss in present discounted value of gross world product under 1.5 ˚C relative to 2 ˚C and positive values indicate a gain in present discounted value of gross world product under 1.5 ˚C relative to 2 ˚C. For the time horizon of 2100 the majority of parameter value combinations indicate that the 1.5 ˚C level of mitigation results in a loss relative to 2 ˚C even under Burke-DICE damages. However, for the time horizon of 2300, all parameter value combinations tested indicate that the 1.5 ˚C level of mitigation results in a gain relative to 2 ˚C under Burke-DICE damages.

and achieving either 1.5 ˚C or 2.0 ˚C require a net sacrifice of GWP, relative to a no-mitigation case, though 2100.

It is relevant to discuss the above results in the context of two related studies [43, 82] that calculate that damages similar to Burke at al. [39] justify the 2.0 ˚C target on purely economic grounds. There are a number of relevant modeling and parameter differences between the present study and these two previous studies that could account for some differences in conclusions (c.f. the present study's Methods section with "Calculation of Damage Costs" in Appendix A2 of Ueckerdt et al. [82] and "Deriving a New Damage Cost Function for DICE" in the Methods section of Glanemann et al. [43]). However, our results are not necessarily inconsistent with either of these studies. A critical point of distinction is that in Figs 12b and 13a, we are discussing outcomes integrated through the time horizon of 2100 while these studies consider the optimal warming level (in 2100) taking into account accrued benefits integrated well beyond the year 2100 (though still time-discounted).

Indeed, when we consider the longer time horizon of 2300 (Figs 12c, 12d and 13b), the distributions of all three levels of mitigation effort (1.5 ˚C, 2.0 ˚C and 3.0 ˚C in 2100) become positive on average for both Default-DICE and Burke-DICE, meaning that under most parameter combinations, the 1.5 ˚C and 2.0 ˚C temperature targets confer net benefits relative to no-mitigation. In particular, The mean of the distribution for Burke-DICE associated with the 1.5 ˚C target is near 2,000 trillion US$ and has a long tail that surpasses 3,500 trillion US$, indicating that the consideration of this longer time horizon and heightened damages, drastically increases the calculated benefits of stringent mitigation. Under Burke-DICE and the 2300 time horizon, nearly every parameter combination tested results in the 1.5 ˚C target producing more accumulated (time-discounted) GWP than the 2.0 ˚C target (Fig 13b, red).

## 4. Discussion and conclusions

The results of this study come with a number of important caveats. First, we use a highly-idealized schematic model of the coupled global climate-economic system (DICE). This simplicity

allows us to make transparent calculations but prevents us from explicitly simulating features of the system that may turn out to be of great importance. This applies to all modules of the DICE model, but it is particularly relevant to the estimation of mitigation costs which we weigh against climate damages in our benefit-cost analyses. Although DICE's mitigation cost calculations produce global values consistent with those produced from processed-based IAMs (Fig 7), they do not explicitly represent individual energy technologies, individual geographical regions, energy systems' inertia, cost declines via learning-by-doing, induced technological change, etc.

Another key limitation is that we have not considered the impact of Paris Accord temperature targets on non-market environmental goods and/or natural capital. The present study has been framed in terms of gross world product so that our results would be comparable to similar calculations from the IPCC [45, 46] and Burke at al. [39]. However, the Paris Accord temperature targets were devised with much more than just economic optimization in mind and thus even if a given level of mitigation effort turns out to be suboptimal from a gross world product standpoint, it could still be optimal in a more holistic framework that places a higher weight on e.g., intangible natural capital like biodiversity. Impacts that are expected to be exacerbated under 2 ˚C of global warming compared to 1.5 ˚C, but are difficult to monetize, include a larger reduction in the strength of the Atlantic Meridional Overturning Circulation [83], a greater amount of ocean acidification [84], increased probability of an ice-free arctic [85], increased frequency of category 4 and 5 tropical cyclones [86], increased habitat loss for insects, vertebrates and plants [87], and increased susceptibility for malaria transmission [88], among many others [10].

Furthermore, even impacts that may be readily monetizable, like the economic effect of sea level rise, will not be captured by historical interannual temperature shocks and are thus not included in Burke at al. [39] damages estimates. On the other hand, the damage calculations of Burke at al. [39] do not anticipate future adaptation and it is controversial as to whether historical temperature variability is an appropriate analog for the economic damages to be expected from sustained climate change. Indeed, it has been suggested that the damages projected by Burke at al. [39] may be substantially overestimated [40, 65, 66] and thus the range between those estimates and the default DICE representation of damages may be able to serve as a rough proxy for the envelope of uncertainty in economic damages from climate change.

Another caveat is that our model not run in optimization mode so the impact of mitigation on gross world product is not at its absolute least-cost limit under conditions of perfect foresight. However, practically speaking, we cannot expect to implement a perfect-foresight, least-cost control rate on $CO_2$ emissions at the global level. Thus, selecting from the spectrum of linearly ramping control rates used here still represents an outcome that is optimistic in terms of maximizing gross world product since the selection would likely avoid many economic inefficiencies associated with real-world implementation of mitigation policy.

Finally, we conduct our analysis using a highly idealized model with a single production function at the global mean level. This means that all of our calculations only apply to the global aggregate and say nothing about the distribution of benefits and costs of mitigation over differing portions of the income distribution [89, 90], or over different geographical locations [91, 92]. In particular, since we conduct our analysis on gross world product and not on the utility of consumption, we do not discount the well-being of future generations (which are wealthier than the present generation in this framework) in the way that they would be in the traditional DICE framework.

The above caveats notwithstanding, our analysis (which is similar to previous studies [21, 32, 33, 43]) is able to reveal some rough first-order insights. First, the incorporation of heightened damages from Burke at al. [39] shifts the year at which a given level of mitigation effort

begins to provide net economic benefit from the 22nd century to well within the 21st century. Thus, under stringent mitigation effort, Default-DICE damages imply, on the global mean, a 21st century entirely characterized by gross world product sacrifice for future generations while Burke-DICE damages imply that net economic benefits will begin to be conferred by the 2070s or 2080s, within the lifetimes of many people alive today (Fig 9).

There are large differences between the two representations of damages in terms of the level of mitigation that results in the maximum present discounted value of gross world product. Under the traditional representation of damages, the 1.5 ˚C and 2 ˚C targets maximize the present discounted value of gross world product only under the combination of long time horizons and low discount rates (beyond 2150, and under 1%/year, Fig 11a). In contrast, under the heightened representation of damages, the 1.5 ˚C and 2 ˚C targets maximize the present discounted value of gross world product starting in the early 21st century with discount rates as high as ~3%/year (Fig 11b).

With the above point in mind, we still choose to highlight calculations using a 3%/year discount rate (because this is the principal discount rate used in Burke at al. [39]) and a time horizon of 2100 (because this is the timeframe used in Burke at al. [39], in IPCC assessments [45, 46], and in Shared Socioeconomic Pathway studies [47]). Under these temporal parameters, we calculate that limiting global warming to 1.5 ˚C tends to result in a net loss in gross world product relative to both the 2 ˚C level of mitigation, as well as relative to the no-mitigation scenario, under both Default-DICE and Burke-DICE representations of damages (blue and green vertical lines in Fig 12a and 12b). Under the Burke-DICE representation of damages, we calculate that achieving the 1.5 ˚C target results in a net loss of approximately 40 trillion US$ in gross world product relative to 2 ˚C (through 2100, 3%/year discount rate). This finding highlights the potentially long payback period associated with the most stringent global warming mitigation targets.

## Acknowledgments

The authors acknowledges Juan Moreno-Cruz, Zeke Hausfather, Steven J. Davis, Fan Tong, and Ken Caldeira for valuable discussions. The Matlab code used for this analysis is available at https://doi.org/10.5281/zenodo.4002104

## Author Contributions

**Conceptualization:** Patrick T. Brown, Harry Saunders.

**Formal analysis:** Patrick T. Brown.

**Project administration:** Patrick T. Brown.

**Writing – original draft:** Patrick T. Brown.

**Writing – review & editing:** Patrick T. Brown, Harry Saunders.

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
