## [Decision Letter · Decision Letter 0]

7 Apr 2020

PONE-D-20-05098

Approximate calculations of the net economic impact of UN global warming mitigation targets under heightened damage estimates

PLOS ONE

Dear Dr. Brown,

Thank you for submitting your manuscript to PLOS ONE. After careful consideration, we feel that it has merit but does not fully meet PLOS ONE’s publication criteria as it currently stands. Therefore, we invite you to submit a revised version of the manuscript that addresses the points raised during the review process.

I have found especially relevant the point raised by the Reviewer#1 about the availability of the code that you have used for your computations. Making the code available is something that we encourage in PLoS ONE and necessary to assure the full reproducibility of published works. Therefore, if you have written sensible parts of code to perform this study, it would be needed that you include them with the manuscript (e.g., linked in a Zenodo repository).

We would appreciate receiving your revised manuscript by May 22 2020 11:59PM. To enhance the reproducibility of your results, we recommend that if applicable you deposit your laboratory protocols in protocols.io, where a protocol can be assigned its own identifier (DOI) such that it can be cited independently in the future. For instructions see: http://journals.plos.org/plosone/s/submission-guidelines#loc-laboratory-protocols

We look forward to receiving your revised manuscript.

Kind regards,

Juan A. Añel, Ph.D.

Academic Editor

PLOS ONE

Journal Requirements:

2. Please note that PLOS ONE has specific guidelines on software sharing (http://journals.plos.org/plosone/s/materials-and-software-sharing#loc-sharing-software) for manuscripts whose main purpose is the description of a new software or software package. In this case, new software must conform to the Open Source Definition (https://opensource.org/docs/osd) and be deposited in an open software archive. Please see http://journals.plos.org/plosone/s/materials-and-software-sharing#loc-depositing-software for more information on depositing your software.

3.  Please ensure that all data sources are clearly listed and referenced in the manuscript, in order to comply with PLOS ONE data sharing policies (https://journals.plos.org/plosone/s/data-availability).

Reviewers' comments:

Reviewer's Responses to Questions

**Comments to the Author**

1. Is the manuscript technically sound, and do the data support the conclusions?

Reviewer #1: Partly

Reviewer #2: Yes

2. Has the statistical analysis been performed appropriately and rigorously? 

Reviewer #1: I Don't Know

Reviewer #2: Yes

3. Have the authors made all data underlying the findings in their manuscript fully available?

Reviewer #1: No

Reviewer #2: No

4. Is the manuscript presented in an intelligible fashion and written in standard English?

Reviewer #1: Yes

Reviewer #2: Yes

5. Review Comments to the Author

Reviewer #1: In their manuscript "Approximate calculations of the net economic impact of UN global warming mitigation targets under heightened damage estimates" the author compares costs of climate mitigation to the climate damages avoided by those measures. He does so by including an empirically derived damage function into the simple integrated assessment model DICE for a set of linear mitigation curves. With the new damage function (compared to the one used in "classical" DICE) the author finds the costs of climate change (measured in global GDP losses) to outweigh the mitigation costs sooner. Net costs in such a comparison, however, are shown to highly depend on the discount rate as well as the time frame considered.

Overall the manuscript underlines the importance of looking at the time scales of mitigation costs and climate damages when comparing the two. It is written in a clear language and well structured in principal. I have, however, some concerns with the methods and the manuscript.

The underlying idea here is to compare climate damages avoided to the necessary mitigation costs. In doing so separately, it is quite similar to the method in Uekerdt et al. [1], who also use Burke et al. 2018 on SSP projections to derive climate damages. They compare these to mitigation costs as per a process-based IAM (REMIND); whereas this manuscript establishes an ensemble of the latter. I very much wonder why [1] get lower optimal temperatures (though when minimizing lost welfare) compared to these results. The author should definitively discuss these differences. Another recent publication with a similar endeavor the author should relate to - at least in the discussions - is Glanemann et al. [2]. They also incorporate Burke et al. 2018 damages into DICE, while separating growth from level effects in the full optimizing mode of DICE. They find the 2°C target to be optimal, even lower temperature targets depending on the SSP.

This manuscript nevertheless adds to the literature by looking at some uncertainties in the mitigation cost curve, but the aforementioned differences to other parts of the literature should be discussed thoroughly.

Unfortunately, many important details of the methodology are missing, which makes it difficult to fully follow the conclusions. Ideally, the author would also provide the underlying source code used.

I was very astonished to see that the author does not seem to use DICE in its standard mode of optimization, but rather uses some of its equations. The most important control variable of DICE, actually, is the savings rate, not the mitigation control rate as stated here. That also the savings rate is derived exogenously, so there is no optimization at all, only became clear to me later throughout the paper. As this is a very important aspect of the methodology here, it should be stated clearly and early.

Overall, it is not clear which variables are exogenous and which are endogenous. Also, though some equations can be found in the respective DICE description papers, this manuscript would much benefit from describing its methodology full for the important variables used here.

This includes:

- How is the capital depreciation rate used in K(t)? K(t) is determined using the SSPs (eq. 7). Is that only to derive the savings rate? What are typical savings rates derived from the SSPs?

- How do beta and xi (eq. 8) evolve? Is that from the original DICE? How are they scaled later in the Monte Carlo simulations? Same factor for the whole time series?

- What are the values for the parameters used here, e.g. gamma in eq. 4, c or theta in eq. 8?

- When is Y(t) used and when GWP(t)? Is the former used as gross GWP and the latter as net GWP?

- For sake of completeness a summarizing equation for the values depicted in the figures (e.g. climate damages, damages avoided) would be very helpful.

- Is only SSP2 used throughout the manuscript after Figure 1?

The method description needs to be much more complete before a publication of this manuscript.

A main point in this manuscript is the ensemble of mitigation scenarios, i.e. time series of mu(t). I wonder how narrowing the assumption of linear mu(t) actually is (assuming a constant reduction rate, i.e. exponential mu(t) seem to be as or even more reasonable). All of these include negative emissions, but for a more complete picture other mu(t) with ambitious mitigation without negative emissions would be highly desirable. Omitting negative emissions in a mu(t) additionally prevents shifting avoided damages into the future and would thus contribute to a more complete picture when looking at the relevant time frames as done in this paper.

Especially for communication in Figure 4 the method would benefit from sampling the mu(t) to have equidistant temperatures in 2100 (as that is the main variable used in the subsequent analysis). Regardless of the sampling used, a full description of the parameters used for that must be given in the method description.

In lines 150-158 the authors states three primary advantages of the method used. I partially agree with the first two, though I think one should note that, since DICE is not used in its optimizing mode, the results of this paper are not easily transferable to "normal" DICE analyses. However, the author claims that the method "removes the philosophical complications associated with converting consumption to human well-being via a utility function" (3rd point). While this is always a problematic issue, it is not avoided here at all. Rather than focusing on well-being via a utility function, this study looks at net production (Y/GWP), which is implicitly assumed to be the relevant measure to be maximized for this cost-benefit analysis (or "to justify the [...] temperature targets" as stated in the abstract). In a revised article this needs to be made more explicit.

Additionally, maximizing Y/GWP in the optimization mode in DICE would essentially mean no consumption. One might thus ask, if sticking to gross production as per the SSP data with a fixed savings rate makes sense when, at the same time, comparing mitigation costs and avoided damages in terms of GWP; in most IAMs this would result in shifting consumption in time additionally avoiding climate damaging production.

On a minor note, I would very much appreciate a short discussion on the concept of the discount rate. Since the costs of climate change go sub-linear in time (Figure 4d), a sufficiently large discount rate (as an exponent for exponentially diminishing sum) will always outweigh them (Figure 7); regardless of the time horizon.

Overall, I believe, when already using an integrated assessment model, the question of cost-benefit would be much better tackled in the full optimizing mode. This would actually embrace the idea of normative assessment in inter-temporal optimization for which such a model is built. Deriving both cost curves independently as done here, totally neglects their interaction (perceived damages in the future).

Nevertheless, after a thorough revision, this publication could definitely contribute to the literature on climate change cost-benefit analyses.

Other comments/questions:

Abstract (lines 13/14)

- I doubt that's a reasonable interpretation of the "break-even year" as used in this manuscript. Here it is the year in which the mitigation costs of that year and those of the damages avoided (for that year!) are the same. This notion totally neglects that the avoided damages of a certain year - due to the inertia in the climate system - result from mitigation efforts before that year. Thus, this definition of "break-even year" is only useful in a technical sense.

Figure 1:

- Please use a legend instead of arrows

- Do you mean "GWP change" (as it is <0, which is meant to be losses)?

Figure 2:

- Are the values on both y-axes "per year" or "per 5 years"?

Figure 3:

- Which discount rate is used here?

- Please use a legend instead the arrow

- "loss" in x-axis label should be removed

Figure 4:

- Please use a legend instead of arrows

- Damages avoided would be an interesting variable

- Please mark y=0 in c & d. Where do damages start?

Make sure earlier that [39 is Burke et al 2018

Line 126: Capital C?

Line 248: What about renewable energies?

Line 558: "[...] it is controversial as to whether historical temperature variability is an appropriate analogue for the economic amages to be expected from sustained climate change": The same argument could be used to argue that the climate damages are underestimated

Line 568 "[...] or over time": Looking at the distribution of the benefits and costs over time is the point of this manuscript, is it not?

References:

[1] Ueckerdt et al. The economically optimal warming limit of the planet. Earth Syst. Dyn., 2019

[2] Glanemann et al. Paris Climate Agreement passes the cost-benefit test. Nat Comm., 2020.

Reviewer #2: I would like to thank the author for a very interesting article on a very topical matter. The paper is extraordinarily well written and very easy to follow. Given the large literature on DICE modelling, I think the paper adds a number of interesting aspects, such as comparing DICE outcomes to the SSPs, yet before this paper is suitable to publish, I would, recommend a number of changes, clarifications and additional discussions to be taken up by the author.

Substantive Comments

1. It seems to me that the author is making a number of highly interesting additions to the DICE modelling framework. Yet overall, the fundamental question the author intends to answer (are the Paris Agreement goals economically beneficial, given new information on the damage relationship?) does not necessitate all these modifications to the DICE set-up. As DICE, in its original form, already equates costs and benefits and hence answers this question, I would be highly interested in the result of DICE with an amended damage relationship, but keeping the emission control rate endogenous, rather than as an exogenous variable derived from the RCPs (see Glanemann et al. 2020, Nature Comms).

2. In line 413, the author contrasts his results to Burke et al (2018), who find achieving 1.5°C over 2°C would result in $36tn economic benefits by 2100, while archiving this target would result in a loss of $9tn in GWP under the author’s framework, which also includes mitigation costs. This would imply mitigation costs of $45tn by 2100 – I would urge the author to discuss whether such a value for mitigation costs seems plausible.

3. The author stresses that the goal of his study is to represent the range of uncertainty between both DICE versions (Line 115). I would suggest representing this intention in the abstract as well, as uncertainty is not currently addressed in the abstract.

4. The author transparently states that in his framework μ(t) must take a linear form. I would appreciate it, if the author would discuss the validity of this assumption based on empirical data (most process-based IAMs tend to suggest an inverse functional relationship being needed/most realistic) and some discussion whether his results would hold under an amended functional form.

5. It remains unclear to me whether the values of a, b, and c were tested for their sensitivity. It seems to me they were not included in the Monte Carlo analysis. As they are crucial values for the subsequent analysis, I would urge the author to discuss whether the choice of values (both individually as well as in relation to each other) would affect the estimation result. Further, I would appreciate a comparison of the study’s values with the empirical evidence of temperature effects on capital depreciation as well as on total factor productivity.

Minor Comments

6. The author should detail the exact data source for Figure 3. I am unclear how the four boxes of for the mitigation costs of RCPs have been calculated and which IAM variable in process-based variables is used for this comparison (I assume the data was taken from the Appendix of reference 65, but more detail would be appreciated).

7. I recognise that the author is following Moore and Diaz in their functional form for equations 5 and 6, but I would appreciate an explanation/discussion on why temperature enters linearly in 5 and in a non-linear form in 6.

8. The goals the article is referring to are not “United Nations” goals. They are specific to certain international treaties and declarations, which have varying membership but are not universally accepted by all member states of the United Nations. In the spirit the author is referring to them, they should be referred to as “Paris Agreement” targets. Similarly, the Paris Agreement did not “affirm” the 2°C target (line 55), as it was enshrined in an internationally legally binding document for the first time then (Copenhagen Accord was not a treaty), while the language was strengthened in the PA to “well-below 2°C” rather than “below 2°C”.

9. For readers unfamiliar with the economics terminology, I would suggest explaining all variables used in equations when they first appear – even if they are as simple as labour, capital and Utility (line 126, the definition is only provided two pages later).

10. In Figure 1, the author could consider presenting SSPs 1,3,4 and 5 in an appendix.

Where possible, Table 1 should feature the Greek letter for the variables in the Monte Carlo analysis as well.

11. In line 296, rather than referring to “comprehensive” IAMs, I would stick to the term used in line 258 “process-based” (alternatively: energy-system models or cost-effectiveness IAMs).

Reference 47 has different formatting to all other references.

12. The author could consider citing Lontzek, TS; Cai, Y; Judd, KL; et al. 2015 when discussing possible non-linear changes in the climate system and their effects on DICE modelling.

6. PLOS authors have the option to publish the peer review history of their article (what does this mean?). If published, this will include your full peer review and any attached files.

Reviewer #1: No

Reviewer #2: No

---

## [Author Response · Author response to Decision Letter 0]

15 May 2020

See attached "Response to Reviewers" document.

---

## [Decision Letter · Decision Letter 1]

1 Jul 2020

PONE-D-20-05098R1

Approximate calculations of the net economic impact of global warming mitigation targets under heightened damage estimates

PLOS ONE

Dear Dr. Brown,

Thank you for submitting your manuscript to PLOS ONE. After careful consideration, we feel that it has merit but does not fully meet PLOS ONE’s publication criteria as it currently stands. Therefore, we invite you to submit a revised version of the manuscript that addresses the points raised during the review process.

I would ask you to check the comments by a new reviewer carefully. Some of them can seem awkward, however, they go to the point, and in my view, there are several issues raised need a good rebuttal. You could consider necessary running further experiments to check the impact of changing parameters.

We look forward to receiving your revised manuscript.

Kind regards,

Juan A. Añel, Ph.D.

Academic Editor

PLOS ONE

Journal Requirements:

1. Please ensure that the original data sources are clear within the manuscript. You may consider specifying these in a separate section. Please also include this information in your Data availability statement.

Reviewers' comments:

Reviewer's Responses to Questions

**Comments to the Author**

1. If the authors have adequately addressed your comments raised in a previous round of review and you feel that this manuscript is now acceptable for publication, you may indicate that here to bypass the “Comments to the Author” section, enter your conflict of interest statement in the “Confidential to Editor” section, and submit your "Accept" recommendation.

Reviewer #2: (No Response)

Reviewer #3: All comments have been addressed

2. Is the manuscript technically sound, and do the data support the conclusions?

Reviewer #2: Yes

Reviewer #3: No

3. Has the statistical analysis been performed appropriately and rigorously? 

Reviewer #2: Yes

Reviewer #3: N/A

4. Have the authors made all data underlying the findings in their manuscript fully available?

Reviewer #2: Yes

Reviewer #3: Yes

5. Is the manuscript presented in an intelligible fashion and written in standard English?

Reviewer #2: Yes

Reviewer #3: No

6. Review Comments to the Author

Reviewer #2: I would like to thank the authors for implementing many of the suggested revisions from my first set of comments. Especially welcome is the revamped methodology section and especially the included schematic figures ensure clarity much better.

Given these changes, I only have a few remaining comments, which are either editorial or mainly concern the paper’s framing. The framing of the paper is ultimately of course up to the authors, but I would encourage the authors to carefully consider my major comments, as I think the possible reception of this paper from certain audiences might deviate from the paper’s actual conclusions.

This overall leads me to conclude that I think this is a highly interesting and well-presented paper which ultimately should be published, but I do believe that the authors should be wary of the reception of their results, which is why I suggest a few small changes to ensure adequate qualifiers are added, where they are necessary.

Major comments:

I understand that the authors have chosen to report a particular specification of discount rate and time preference as their main result in order to compare their results to Burke et al’s number of US$36tn. However, considering the major implications their claims could have regarding the 1.5°C and 2°C targets independent of Burke et al’s result, I would urge the authors to qualify their 2100 main result with reference to the considerable sensitivity that is highlighted in their paper. The authors themselves argue that the sign of the 2100 result is sensitive to variation in their parameters and that across reasonable parameters the negative result would not hold for e.g. 90 or 95 per cent of the analysed parameters (the authors list that this is the case for 70% of the 2°C result and 80% of the runs for the 1.5°C result in their Burke-DICE) – in contrast to the 2300 result where the sign of the estimated effects seems to be clear. I would suggest to the authors to qualify their main result for 2100 in the appropriate places (e.g. abstract, conclusion, etc.) with a reference to a certain sensitivity to the parameters chosen.

In a similar vein, I would argue the authors should at least mention that their calculations are not cost optimal pathways in the abstract and the conclusions.

I thank the authors for adding the reference that now clarifies that US$ 45tn is a reasonable estimate for the mitigation costs; at least according to the existing literature.

Perhaps the authors could add whether this is also the number that their model uses for the mitigation costs based on the SSPs. Currently, the authors simply compare their overall GWP loss to Burke et al’s damage statement, which implies the US$45tn mitigation cost. In effect, I would be interested to see how close the authors parameterization of the damages gets to Burke et al’s original number.

Minor comments:

I would appreciate if the authors could restate in one sentence or so how Burke et al 2018 have arrived at their US$36tn figure so as to make the comparison easier (around line 480).

It is not entirely clear which study the authors are basing their damage parameters from. In a number of places, they refer to Burke et al 2015 and in others to Burke et al 2018. I would guess that their reference to Burke et al 2015 is wrong e.g. in Fig 3 caption, line 237, etc.

In the legend for Figure 1, the authors forgot the parameter h

I thank the authors for making their replication code available. Before publication, however, they should ensure to add replication instructions to the GitHub repository.

Reference 87 seems to have a formatting issue regarding the names of the authors in the bibliography.

Reviewer #3: I did not review the previous version of the paper. The authors worked hard to revise the paper, but more work is needed. The language is awkward, the authors mess up their welfare theory, and the rely on a wrong model (see below).

Furthermore, there is little new in here. The authors refer to a series of papers that do very similar things.

"Traditional DICE" Traditional? Please consult a dictionary.

"we find it simplifying" ugly language

"to eliminate the social welfare function [...] so that the specific values of the normative parameters are not latently affecting our GWP calculations" what is "latently affecting"? more importantly, you did not remove the normative parameters, you set the rate of risk aversion to zero. You may want to reread Bernoulli's work on this.

"It eliminates the latent influence of normative parameters" No, it does not. It just replaces one set of normative parameters with another. The proposed method is not at all new. In the literature, FUND and PAGE have been running in this mode for decades.

"pretext" see above remark about a dictionary

"suggest" is the right word -- there is little evidence that climate change affects the growth rate of output rather than its level -- I do not need to point you to the relevant papers, which are in your bibliography.

The Burke model is not credible. It is by now common knowledge that there econometrics are wrong. They claim to regress a stationary variable (the change in log per capita income) on a non-stationary one (temperature). In fact, the regress income growth on the cointegrating vector of temperature and year dummies. This works in-sample, although their parameters are biased and their standard errors meaningless. Out-of-sample, the model goes quickly off the rails.

7. PLOS authors have the option to publish the peer review history of their article (what does this mean?). If published, this will include your full peer review and any attached files.

Reviewer #2: No

Reviewer #3: No

---

## [Decision Letter · Decision Letter 2]

28 Jul 2020

PONE-D-20-05098R2

Approximate calculations of the net economic impact of global warming mitigation targets under heightened damage estimates

PLOS ONE

Dear Dr. Brown,

Thank you for submitting your manuscript to PLOS ONE. After careful consideration, we feel that it has merit but does not fully meet PLOS ONE’s publication criteria as it currently stands. Therefore, we invite you to submit a revised version of the manuscript that addresses the points raised during the review process.

As you can see from their comments, one of the reviewers is not convinced by the modifications that you have performed. It seems hard that you reach an agreement, but I think that you can reach an acceptable version for both parts.

I think that the comment by the reviewer about toning down the statements included in the paper is right, and you should adopt it. Also, I would encourage you to at least discuss better the limitations of the Burke model in the text and their potential impact on your results. Maybe including a brief specific subsection on this would be a possibility.

We look forward to receiving your revised manuscript.

Kind regards,

Juan A. Añel, Ph.D.

Academic Editor

PLOS ONE

Journal Requirements:

1) Please ensure that the original data sources are clear within the manuscript. You may consider specifying these in a separate section. Please also include this information in your Data availability statement.

Reviewers' comments:

Reviewer's Responses to Questions

**Comments to the Author**

1. If the authors have adequately addressed your comments raised in a previous round of review and you feel that this manuscript is now acceptable for publication, you may indicate that here to bypass the “Comments to the Author” section, enter your conflict of interest statement in the “Confidential to Editor” section, and submit your "Accept" recommendation.

Reviewer #3: (No Response)

2. Is the manuscript technically sound, and do the data support the conclusions?

Reviewer #3: No

3. Has the statistical analysis been performed appropriately and rigorously? 

Reviewer #3: N/A

4. Have the authors made all data underlying the findings in their manuscript fully available?

Reviewer #3: Yes

5. Is the manuscript presented in an intelligible fashion and written in standard English?

Reviewer #3: Yes

6. Review Comments to the Author

Reviewer #3: Dismissing comments will get you nowhere.

Moyer et al., Moore and Diaz, Dietz and Stern and Ricke et al. have all done very similar things. You need to tone down your claim of novelty.

The Burke model is wrong. It may be frequently cited, but that does not make it right. You will need to defend your specification. Hiding behind authority, or in this case faux-authority is not an option.

7. PLOS authors have the option to publish the peer review history of their article (what does this mean?). If published, this will include your full peer review and any attached files.

Reviewer #3: No

---

## [Author Response · Author response to Decision Letter 2]

31 Jul 2020

See uploaded "response to reviewers" document.

---

## [Decision Letter · Decision Letter 3]

10 Aug 2020

PONE-D-20-05098R3

Approximate calculations of the net economic impact of global warming mitigation targets under heightened damage estimates

PLOS ONE

Dear Dr. Brown,

The reviewer#3 has provided additional comments on your manuscript. I acknowledge that they are exigent. However, I feel that we are moving quickly in the review process and that your paper is being hugely improved. Therefore, I would like to ask you to make an extra effort to address their concerns. You can see their comments below.

We look forward to receiving your revised manuscript.

Kind regards,

Juan A. Añel, Ph.D.

Academic Editor

PLOS ONE

Reviewers' comments:

Reviewer's Responses to Questions

**Comments to the Author**

1. If the authors have adequately addressed your comments raised in a previous round of review and you feel that this manuscript is now acceptable for publication, you may indicate that here to bypass the “Comments to the Author” section, enter your conflict of interest statement in the “Confidential to Editor” section, and submit your "Accept" recommendation.

Reviewer #3: (No Response)

2. Is the manuscript technically sound, and do the data support the conclusions?

Reviewer #3: Partly

3. Has the statistical analysis been performed appropriately and rigorously? 

Reviewer #3: N/A

4. Have the authors made all data underlying the findings in their manuscript fully available?

Reviewer #3: Yes

5. Is the manuscript presented in an intelligible fashion and written in standard English?

Reviewer #3: Yes

6. Review Comments to the Author

Reviewer #3: We're slowly getting there.

You now admit that Burke might be wrong, but continue to refuse to spell out what's wrong and what's right with that paper.

But now that you've admitted the mistake, the paper has a serious flaw: You compare Nordhaus-DICE to Burke-DICE. The comparison masks two changes at once: You switch from damages-in-level to damages-in-growth and from credible-damage-estimates to disputed-damage-estimates. (Disputed is a euphemism for not-credible.) You therefore need to add an intermediate case, with damages-in-growth as estimated by Fankhauser, Dell, Letta or Kahn.

The reference to Kahn is all messed up.

7. PLOS authors have the option to publish the peer review history of their article (what does this mean?). If published, this will include your full peer review and any attached files.

Reviewer #3: No

---

## [Editor Report · Decision Letter 4]

26 Aug 2020

PONE-D-20-05098R4

Approximate calculations of the net economic impact of global warming mitigation targets under heightened damage estimates

PLOS ONE

Dear Dr. Brown,

Thank you for submitting your manuscript to PLOS ONE. After careful consideration, we feel that it has merit but does not fully meet PLOS ONE’s publication criteria as it currently stands. Therefore, we invite you to submit a revised version of the manuscript that addresses the points raised during the review process. See next. Please, be aware that in PLoS ONE we do not have a proofs stage before publication and this is why we can ask for some changes in format before accepting a manuscript for publication.

Between numerals and units, a blank space is mandatory (e.g., 19 °C). Please, correct it along the manuscript.

line 39 - remove Fahrenheit degrees. Fahrenheit units do not provide any relevant additional information.

line 65 - "all of which are used by the U.S. government to estimate the global social cost of carbon." This sentence is anecdotal and unnecessary. Please, remove it.

line 89 - The sentence reads: "In particular, the results of [38] and [36, 37] suggest substantial effects...". In PLoS ONE, we use a numeric citation style. For the sake of correct grammar, this entails writing in style different to when using the 'surname et al.' approach. For example, your sentence here should read something like "In particular, previous results [36-38] suggest substantial effects...". Please, correct it along the manuscript.

line 224: "sea level"

Please, use ZENODO as the repository for your code. Zenodo is widely used and recommended.

https://journals.plos.org/plosone/s/recommended-repositories

There are significant concerns about the reliability of Github for scientific purposes

https://www.nature.com/articles/d41586-018-05426-0

and to duplicate your code in Zenodo is straightforward:

https://guides.github.com/activities/citable-code/

We look forward to receiving your revised manuscript.

Kind regards,

Juan A. Añel, Ph.D.

Academic Editor

PLOS ONE

---

## [Editor Report · Decision Letter 5]

9 Sep 2020

Approximate Calculations of the Net Economic Impact of Global Warming Mitigation Targets Under Heightened Damage Estimates

PONE-D-20-05098R5

Dear Dr. Brown,

We’re pleased to inform you that your manuscript has been judged scientifically suitable for publication and will be formally accepted for publication once it meets all outstanding technical requirements.

Kind regards,

Juan A. Añel, Ph.D.

Section Editor

PLOS ONE
---

## [Editor Report · Acceptance letter]

15 Sep 2020

PONE-D-20-05098R5

Approximate Calculations of the Net Economic Impact of Global Warming Mitigation Targets Under Heightened Damage Estimates

Dear Dr. Brown:

I'm pleased to inform you that your manuscript has been deemed suitable for publication in PLOS ONE. Congratulations! Your manuscript is now with our production department.

Kind regards,

on behalf of

Dr. Juan A. Añel 

Section Editor

PLOS ONE